# Rethinking Neural Network Learning Rates: A Stackelberg Perspective

**Sihan Zeng** [1]  **Sujay Bhatt** [1]  **Sumitra Ganesh** [1]

## Abstract

Neural networks are typically trained with a single learning rate across all layers. While recent empirical evidence suggests that assigning layer-specific learning rates can accelerate training, a principled understanding of the conditions and mechanisms under which non-uniform learning rates are beneficial remains limited. In this work, we investigate non-uniform learning rates through the lens of Stackelberg optimization. Specifically, we demonstrate that training neural networks with a smaller learning rate for the body layers and a larger learning rate for the final layer can be interpreted as a two-time-scale alternating gradient descent algorithm applied to a Stackelberg reformulation of the original objective. We establish finite-time convergence guarantees for the algorithm under broad conditions that accommodate constraint sets and non-smooth activation functions. Beyond convergence, we identify two mechanisms by which non-uniform learning rates can outperform uniform learning rates: (i) we show that certain problem instances induce a Stackelberg objective with stronger optimization structure than the original objective, yielding faster convergence to globally optimal solutions, (ii) our numerical analysis reveals that the Stackelberg objective can exhibit substantially sharper local curvature, especially in early training, which leads to more informative gradients and learning acceleration. Experiments in supervised learning and reinforcement learning support our findings.

## 1. Introduction

The optimization landscape of modern neural networks is fundamentally shaped by the interplay between representation learning and task-specific prediction. In state-of-the-art architectures, this relationship is typically structured as a deep non-linear feature extractor (the "body") followed by a linear transformation (the "head"). This partitioning is ubiquitous across domains: in supervised learning, the head provides the final logit scores or regression values; in reinforcement learning, the head serves as a linear function approximator operating on the non-linear state features extracted by the body.

When a network is partitioned in this manner, the choice of learning rates (a.k.a step sizes) is a primary determinant of stability and convergence speed (Borkar, 2008). While practitioners often default to uniform learning rates, such an approach may lead to inefficiency or sub-optimality. A number of recent works have reported empirical evidence suggesting that updating different layers at different rates can, in some cases, accelerate the optimization of the loss function (Marion & Berthier, 2023; Shin et al., 2024; Martinez et al., 2025; Galashov et al., 2025; Barboni et al., 2025; Behrouz et al., 2026). An explanation for this observation is that non-uniform learning rates induce a time-scale separation between representation learning in the body layers and task-specific adaptation in the final layer. This allows one component of the network to rapidly adapt to a relatively stable target set by the other, thereby stabilizing the learning dynamics and avoiding unstable limit cycles (Borkar, 2008).

While the explanation is intuitively appealing, a principled mathematical analysis of when and how non-uniform learning rates accelerate optimization from a finite-time and optimization geometry viewpoint is limited.w We believe that understanding the learning rate interplay in modern neural networks is paramount: it moves the field beyond empirical trial-and-error towards a rigorous characterization of the coupled dynamics between feature extraction and task-specific optimization.

**Main Contributions**

- We demonstrate that neural network training with non-uniform learning rates, across regression, classification, and reinforcement learning paradigms, can be viewed as a two-time-scale alternating stochastic gradient descent (SGD) algorithm applied to a Stackelberg reformulation of the original training objective. Leveraging and extending the framework of two-time-scale stochastic approximation (SA), we establish that this alternating SGD

[1]JPMorgan AI Research, United States. Correspondence to: Sihan Zeng <sihan.zeng@jpmchase.com>.

approach is globally convergent to first-order stationary points–even under the challenges of non-smooth activation functions (e.g., ReLU) and parameter constraint sets. Specifically, we derive a general convergence rate of $\widetilde{\mathcal{O}}(k^{-2/5})$, which improves to a faster rate of $\widetilde{\mathcal{O}}(k^{-2/3})$ under the assumption of a strongly convex Stackelberg objective. Our results demonstrate that non-uniform learning rates are a mathematically principled necessity arising from the structural properties of deep learning objectives, rather than a mere empirical heuristic.

- We demonstrate that non-uniform learning rates can induce a Stackelberg reformulation that is strongly convex, even when the joint objective is non-convex. This structural emergence enables global convergence and accelerated rates unattainable under uniform updates. Our findings provide a rigorous theoretical justification for the empirical success of multi-timescale training, characterizing it as a mechanism for exposing more tractable optimization landscapes within complex neural models.

- In cases where convexity is absent, we identify an additional acceleration mechanism: our numerical analysis and optimization landscape visualization reveal that the Stackelberg objective can exhibit sharper curvature. The curvature enhancement provides stronger gradient signals for the body layers, facilitating a more rapid escape from spurious sub-optimal regions and improved convergence speed in the transient regime.

- We complement our analysis with experiments across a range of regression, classification, and reinforcement learning benchmarks, where non-uniform learning rates consistently perform better than, or on par with, standard single-learning-rate training. This provides empirical support for our theoretical insights.

- As a further contribution, we show that our framework, when specialized to temporal difference learning with gradient correction (TDC), yields the first provably convergent variant of this algorithm under neural network function approximation with closed-form updates. This result is obtained by recognizing the body layers as a representation learner that evolves on a slower time scale, while the final layer solves the TDC objective under linear function approximation, and is potentially of independent interest beyond the main focus of this paper.

### 1.1. Related Work

Our paper studies the structure and learning dynamics of neural networks from a Stackelberg perspective, building on and extending analytical tools from two-time-scale SA.

**Neural Network Learning Dynamics.** The classic, widely held belief is that gradients in shallow (early) layers tend to have smaller magnitudes than those in deeper (later) layers, a phenomenon referred to as "vanishing gradient" which slows down learning in the shallow layers. To address the issue, Singh et al. (2015); Ginsburg et al. (2019) introduce layer-wise learning rate adaptation based on curvature or second moment information, leading to accelerated learning in practice. Recent works, however, challenge this belief – through comprehensive numerical analysis, Chen et al. (2023) discover that, contrary to the vanishing-gradient intuition, shallow layers often converge faster than deeper layers due to a smoother optimization landscape.

Galashov et al. (2025) propose closed-form or near-instantaneous optimization of the last layer for regression objectives, highlighting the benefit of fast final-layer adaptation in accelerating training. However, a closed-form solution for final-layer weights is usually only available under $\ell_2$-norm regression losses, making their approach inapplicable in more complex settings. Marion & Berthier (2023) analyze two-layer neural networks training and show that when the output layer is updated sufficiently faster than the body layer, the training dynamics can be rigorously characterized and asymptotic convergence guarantees can be established via an ODE approach, under assumptions of sigmoid activation function and sufficiently large neural network width. In comparison, we establish finite-time convergence under general, possibly non-differentiable activation functions, without restrictions on the network depth and width.

Another notable line of work (You et al., 2017; 2020; Huo et al., 2020; Liu et al., 2024; Hao et al., 2025) study layer-wise and coordinate-wise learning rate adjustment from a different perspective, motivated by improving optimization efficiency through either gradient normalization or noise reduction in stochastic gradients. These works support the broader observation that modifying per-layer learning dynamics can significantly impact convergence behavior, although they do not explicitly connect such schemes to a Stackelberg or two-time-scale formalism. These approaches also rely on the choice of a base learning rate, which is typically set equal across layers. Our work can be complementary to such studies in the sense that it raises the question of whether the further optimization efficiency may arise from setting non-uniform base learning rates across layers in combination with gradient normalization and noise reduction mechanisms.

It is also worth noting the works that study neural network training dynamics through the spectral evolution of the weight matrices (Martin & Mahoney, 2021; Olsen et al., 2025). Their analysis characterizes how optimization induces structured spectral behavior during training and provides a complementary perspective on neural network optimization.

**Bi-Level Optimization and Two-Time-Scale SA.** Bi-level optimization studies hierarchical problems in which an upper-level objective depends on the solution of a lower-level optimization program. It has a close mathematical connection to two-time-scale SA, which aims to find the fixed point of a coupled system of two operators. Gradient-based algorithms for bi-level optimization typically take a two-time-scale SA approach (Kwon et al., 2023; Hong et al., 2023; Zeng et al., 2024; Chen et al., 2024): the lower-level variable is updated on a faster time scale for it to track a best-response to the slowly updated upper-level variable, which facilitates gradient estimation of the upper-level objective. Existing analyses largely focus on unconstrained and fully differentiable objectives, whereas we consider constrained and potentially non-differentiable settings to model neural network training with general activation functions such as ReLU, and establish convergence guarantees of comparable order to those obtained in smooth, unconstrained bi-level optimization.

## 2. Formulation

Let $\Theta \subseteq \mathbb{R}^d$ denote the parameter space of a neural network and $\Xi$ be a statistical state space. We define the training objective as the minimization of the risk function $f$:

$$\theta^\star = \operatorname*{argmin}_{\theta \in \Theta} f(\theta) \triangleq \mathbb{E}_{\xi \sim \mathcal{D}}[\ell(\theta, \xi)], \tag{1}$$

where $\xi \in \Xi$ is a sample from an unknown distribution $\mathcal{D}$, and $\ell : \Theta \times \Xi \to \mathbb{R}$ is a proper, Lipschitz continuous loss.

The formulation in (1) is general enough to encompass a wide range of learning paradigms. In **supervised learning**, we have $\xi = (x, y)$, where $x$ denotes the feature vector and $y$ denotes the ground-truth label. In **temporal-difference learning with correction (TDC)** for policy evaluation, $\xi = (s, a, s')$ corresponds to a tuple of state, action, and next state, and $\ell$ represents the mean squared projected Bellman error. We discuss these applications in detail in Section 5.

### 2.1. Functional Form and Parameter Partitioning

To explicitly see how $f$ arises from the network architecture, we partition the parameters as $\theta = (M, w) \in \mathcal{M} \times \mathcal{W}$. Here, $\mathcal{M}$ denotes the parameter space of the hidden layers (the "body") and $\mathcal{W}$ represents that of the final linear layer (the "head"). There is an activation function $\phi : \mathbb{R} \to \mathbb{R}$ after each body layer, applicable element-wise to vectors and matrices. When the network input is $\xi$, we use $\psi(M; \xi)$ to denote the feature representation produced by the body, which leads to the following network output

$$h(\xi; M, w) = w^\top \psi(M; \xi).$$

Consequently, we can express the sample loss in (1) as

$$\ell(M, w, \xi) = \mathcal{L}\big(h(\xi; M, w), g(\xi)\big) = \mathcal{L}\big(w^\top \psi(M; \xi), g(\xi)\big),$$

where $\mathcal{L}$ is a convex criterion and $g(\xi)$ is a sample-dependent target (e.g., the label $y$ in supervised learning, or the Bellman target in TDC). We assume $\mathcal{M}$ and $\mathcal{W}$ are convex, compact sets and the activation function is Lipschitz continuous but not necessarily differentiable (e.g., ReLU).

### 2.2. Differentiability and the Clarke Subdifferential

Note that $\ell$ remains differentiable with respect to $w$ regardless of the differentiability of the activation function. By the chain rule, we have $\nabla_w \ell(M, w, \xi) = \frac{\partial \mathcal{L}}{\partial h} \cdot \psi(M; \xi)$. Since the potentially non-differentiable activation function is isolated within $\psi(M; \xi)$, it acts as a constant relative to $w$.

Conversely, $\ell$ is not necessarily differentiable with respect to $M$. We use $G_M$ to denote a subgradient sampling operator such that $G_M(M, w, \xi) \in \partial_M \ell(M, w, \xi)$, where $\partial$ denotes the Clarke subdifferential. Since $\ell$ is Lipschitz in $M$, the Clarke subdifferential is guaranteed to be a non-empty, convex, and compact set (Clarke, 1990)[Proposition 2.1.2].

### 2.3. Non-Uniform Learning Rates

In practice, neural network training with non-uniform learning rates follows the update rules below

$$
\begin{aligned}
M_{k+1} &= \operatorname{proj}_{\mathcal{M}} \Big( M_k - \alpha_k G_M(M_k, w_k, \xi_k) \Big), \\
w_{k+1} &= \operatorname{proj}_{\mathcal{W}} \Big( w_k - \beta_k \nabla_w \ell(M_k, w_k, \xi_k) \Big),
\end{aligned}
\tag{2}
$$

where $\xi_k$ is drawn i.i.d. from $\mathcal{D}$, and the learning rates $\alpha_k, \beta_k$ decay polynomially with $k$ at different rates.

*If non-uniform learning rates were the answer, what would the question be?* While historically non-uniform learning rates are used heuristically without a clear mathematical justification, we show in the next section that under proper choices of $\alpha_k, \beta_k$, the algorithm (2) finds a solution of the following Stackelberg optimization problem

$$
\begin{aligned}
M^\star = \operatorname*{argmin}_{M \in \mathcal{M}} \quad & \Phi(M) \triangleq f(M, w^\star(M)) \\
\text{s.t.} \quad & w^\star(M) = \operatorname*{argmin}_{w \in \mathcal{W}} f(M, w).
\end{aligned}
\tag{3}
$$

This objective formalizes a hierarchical view where the body-layer parameters $M$ are optimized while being aware that the final-layer parameters $w$ adapt optimally to the representation induced by $M$. Note that (1) and (3) are equivalent in the sense that they share the same set of globally optimal solutions and every solution of one problem has a corresponding solution for the other. However, their underlying landscapes and structure may be widely different – a subject that we further study and explain in Section 4.

# 3. Non-Uniform Learning Rates Optimize Stackelberg Objective

In this section, we establish the convergence of (2) to (approximate) solutions of the Stackelberg objective (3). We start by introducing the technical assumptions.

**Assumption 3.1.** There exists a constant $\lambda > 0$ such that for all $M \in \mathcal{M}$ and $w, w' \in \mathcal{W}$,

$$f(M, w') \geq f(M, w) + \langle \nabla_w f(M, w), w' - w \rangle + \frac{\lambda}{2} \|w' - w\|^2.$$

The first assumption requires that, for any fixed $M$, the objective $f$ is strongly convex with respect to the final-layer parameters $w$. The assumption is mild and commonly satisfied in practice. In supervised learning problems such as squared-loss regression or logistic regression, strong convexity in $w$ holds naturally when the feature representation induced by $M$ is full rank. Even when $f(M, \cdot)$ is only convex (which always holds in these problems free of assumptions), strong convexity can be enforced by adding a small $\ell_2$-regularization to the final layer with weight $\lambda/2$.

Assumption 3.1 guarantees that the final-layer best-response $w^\star(M)$ is unique. If $f$ is differentiable in $M$, Danskin's theorem (Bertsekas, 1997) states that $\Phi$ is everywhere differentiable and that the gradient of $\Phi$ can be expressed as

$$\nabla \Phi(M) = \nabla_M f(M, w^\star(M)).$$

However, due to the non-differentiability of $f$ in $M$, we cannot invoke Danskin's theorem in its standard form. We derive the following lemma on the subdifferential of $\Phi$.

**Lemma 3.2.** *Under Assumption 3.1, we have*

$$\partial \Phi(M) = \partial_M f(M, w^\star(M)), \quad \forall M \in \mathcal{M}. \qquad (4)$$

Lemma 3.2 reveals a simple but important interpretation of the algorithm in (2): the upper-level variable $M$ is updated along a (biased) stochastic subgradient of the Stackelberg objective, computed using the current iterate $w_k$, whereas $w_k$ is updated along $\nabla_w f(M_k, w_k)$ towards the optimizer $w^\star(M_k)$ to reduce the upper-level subgradient bias.

**Assumption 3.3.** There exists a constant $\rho \in (0, \infty)$ such that the function $\Phi$ is $\rho$-weakly convex, implying for all $M, M'$ and $\nu \in \partial \Phi(M)$

$$\Phi(M') \geq \Phi(M) + \langle \nu, M' - M \rangle - \frac{\rho}{2} \|M - M'\|^2. \qquad (5)$$

Our second assumption is on the weak convexity of $\Phi$, a notion that relaxes the smoothness condition when $\Phi$ is neither convex nor differentiable. Weak convexity is a minimal regularity condition that ensures $\Phi$ behaves like a smooth function. If the activation function is differentiable and smooth,

it is straightforward to show that $\Phi$ is smooth, which implies that $\Phi$ is weakly convex with the same constant. When the activation function is non-differentiable (e.g. ReLU, LeakyReLU) and $f$ is not jointly convex in $(M, w)$, we may still show that $\Phi$ satisfies Assumption 3.3 (see Section 5.1).

By definition, the stochastic (sub)gradients used in (2) are unbiased, i.e. $\mathbb{E}_{\xi \sim \mathcal{D}}[G_M(M, w, \xi)] \in \partial_M f(M, w)$ and $\mathbb{E}_{\xi \sim \mathcal{D}}[\nabla_w \ell(M, w, \xi)] = \nabla_w f(M, w)$. Our next assumption imposes that their variances are bounded.

**Assumption 3.4.** For any $M, w$, there exists $\nu \in \partial_M f(M, w)$ and a constant $\sigma \in (0, \infty)$ such that

$$\mathbb{E}_{\xi \sim \mathcal{D}}[\|G_M(M, w, \xi) - \nu\|^2] \leq \sigma^2,$$
$$\mathbb{E}_{\xi \sim \mathcal{D}}[\|\nabla_w \ell(M, w, \xi) - \nabla_w f(M, w)\|^2] \leq \sigma^2.$$

Our final assumption below is on the Lipschitz continuity of the objective and its (sub)gradients. This is also a mild assumption and can be shown to hold in the problems discussed in Section 5. Note that we do not assume $G_M$ is Lipschitz in $M$, which is much stronger and unlikely to hold when $f$ is not differentiable in $M$.

**Assumption 3.5.** There exists a constant $L < \infty$ such that for all $M, M', w, w'$

$$|f(M, w) - f(M', w')| \leq L(\|M - M'\| + \|w - w'\|),$$
$$\|G_M(M, w, \xi) - G_M(M, w', \xi)\| \leq L\|w - w'\|,$$
$$\|\nabla_w \ell(M, w, \xi) - \nabla_w \ell(M', w', \xi)\| \leq$$
$$L(\|M - M'\| + \|w - w'\|).$$

Our first main result establishes convergence of (2) in the non-convex and non-smooth setting. Since $\Phi$ may not be differentiable, we measure stationarity using a smooth surrogate defined via the Moreau envelope. For any $\hat{\rho} > \rho$, we define the Moreau envelope of $\Phi$ over the constraint set $\mathcal{M}$ as

$$\Phi_{1/\hat{\rho}}(M) \triangleq \min_{M'} \left\{ \Phi(M') + \mathbf{1}_{\mathcal{M}}(M') + \frac{\hat{\rho}}{2} \|M - M'\|^2 \right\},$$

where $\mathbf{1}_{\mathcal{M}}$ denotes the indicator function of the closed convex set $\mathcal{M}$. The Moreau envelope is everywhere differentiable and provides a canonical notion of approximate stationarity for weakly convex functions.

**Theorem 3.6.** *Define* $\hat{M}_k = \operatorname{argmin}_{M'} \{ \Phi(M') + \mathbf{1}_{\mathcal{M}}(M') + \frac{\hat{\rho}}{2} \|M_k - M'\|^2 \}$, *where* $\{M_k\}$ *are the iterates generated by* (2) *under the step sizes*

$$\alpha_k = \frac{\alpha_0}{(k+1)^{3/5}}, \quad \beta_k = \frac{\beta_0}{(k+1)^{2/5}}, \qquad (6)$$

*with* $\alpha_0, \beta_0$ *satisfying* $\alpha_0 \leq \beta_0 \leq 1$ *and* $\beta_0 \leq \min\{\frac{\lambda}{2L^2}, \frac{2}{\lambda}\}$. *Under Assumptions 3.1-3.5, we have for all* $k \geq 0$

$$\min_{t < k} \mathbb{E}\left[ \left( \operatorname{dist}(0, \partial \Phi(\hat{M}_t) + N_{\mathcal{M}}(\hat{M}_t)) \right)^2 \right] \leq \mathcal{O}\left( \frac{1}{(k+1)^{\frac{2}{5}}} \right),$$

*where $N_{\mathcal{M}}(\hat{M}_t)$ denotes the normal cone of $\mathcal{M}$ at $\hat{M}_t$.*

With the exact dependency on structural parameters deferred to Appendix B, Theorem 3.6 states that, under appropriate non-uniform learning rates – where the last layer is updated properly faster than the body layers – the iterates of (2) converge to a first-order stationary point of a smoothed proxy of the Stackelberg objective, with a rate of $\widetilde{\mathcal{O}}(k^{-2/5})$. This matches the best known rate of two-time-scale stochastic approximation in the smooth, non-convex setting (Hong et al., 2023). We note that the rate can be improved to $\widetilde{\mathcal{O}}(k^{-1/2})$ under the smoothness of $w^\star$, via an innovative lower-level residual decomposition scheme introduced in Shen & Chen (2022). However, $w^\star$ is generally non-smooth in our case.

We next turn to a more structured regime where the Stackelberg objective $\Phi$ is strongly convex. While strong convexity does not hold in general, it can arise in important special cases, which we discuss in Section 4.

**Theorem 3.7.** *Consider the iterates of* (2) *under step sizes*

$$\alpha_k = \frac{\alpha_0}{k+h+1}, \quad \beta_k = \frac{\beta_0}{(k+h+1)^{2/3}}, \quad (7)$$

*where $\alpha_0, \beta_0, h$ are selected such that $\alpha_k \leq \beta_k \leq 1$, $\beta_k \leq \min\{\frac{\lambda}{2L^2}, \frac{2}{\lambda}\}$, $\frac{\alpha_0}{\beta_0} \leq \frac{2\lambda}{\lambda_\Phi}$, and $\alpha_0 \geq \frac{8}{\lambda_\Phi}$. Suppose that Assumptions 3.1-3.5 hold, and that $\Phi$ is $\lambda_\Phi$-strongly convex on $\mathcal{M}$, i.e. for all $M, M'$ and $\nu \in \partial\Phi(M)$*

$$\Phi(M') \geq \Phi(M) + \langle \nu, M' - M \rangle + \frac{\lambda_\Phi}{2}\|M' - M\|^2.$$

*Then, we have for all $k \geq 0$*

$$\mathbb{E}[\|M_k - M^\star\|^2] \leq \mathcal{O}\left(\frac{1}{(k+h+1)^{\frac{2}{3}}}\right).$$

Theorem 3.7 establishes the algorithm convergence with rate $\mathcal{O}(k^{-2/3})$ to a globally optimal solution under strong convexity. This rate again mirrors that in the smooth setting (Hong et al., 2023). Theorems 3.6 and 3.7 together confirm that non-uniform learning rates are not merely a heuristic tuning strategy, but rather solve a well-defined Stackelberg optimization problem which shares the same set of optimizers as the original objective in (1).

## 4. What Enables Faster Convergence

While Section 3 establishes convergence of stochastic gradient descent under non-uniform learning rates, it does not yet explain why such asymmetry can lead to faster training. In this section, we identify two factors possibly responsible for this acceleration, supported by mathematical derivations and numerical analysis of the optimization landscape.

**Factor I: Stronger Structure of the Reduced Objective.** A key effect of non-uniform learning rates is that the body layers effectively optimize the reduced Stackelberg objective $\Phi$ rather than the joint objective evaluated at the current iterate $f(\cdot, w_k)$. This change in objective can fundamentally alter the optimization geometry. Even when the joint objective $f(M, w)$ is non-convex, the reduced objective $\Phi(M)$ may satisfy substantially more favorable properties. The following lemma formalizes the existence of such regimes.

**Lemma 4.1** (Convexification via Stackelberg Reduction). *There exist objectives $f : \mathcal{M} \times \mathcal{W} \to \mathbb{R}$ that are not jointly convex in $(M, w)$, whose reduced objective $\Phi(M) = f(M, w^\star(M))$ is strongly convex with respect to $M$ on $\mathcal{M}$.*

Lemma 4.1 shows that the strong convexity of the Stackelberg objective can arise in certain practical problem instances. We acknowledge that, given a general non-convex objective, it may be challenging to test whether the Stackelberg reduction yields strong convexity. Nevertheless, the lemma highlights an important structural possibility and a potential source of acceleration: the standard SGD analysis states that the iterates $(M_k, w_k)$ under a single learning rate only converges to a stationary point of the joint objective $f$ with rate $\widetilde{\mathcal{O}}(k^{-1/2})$, whereas strong convexity of the reduced objective $\Phi$ allows us to invoke Theorem 3.7 and conclude that $M_k$ generated by (2) converges to $M^\star$ with rate $\mathcal{O}(k^{-2/3})$. This represents a quantitative acceleration in convergence rate, as well as an improvement of the convergence notion, from local stationarity to global optimality.

**Factor II: Sharper Curvature in Transient Time.** While the Stackelberg reduction does not generally induce better global structure such as strong convexity, we observe that it can still accelerate convergence by improving the local curvature with respect to the body-layer weights. This effect is illustrated in Figure 1 through a three-dimensional visualization of the optimization landscape.

As an illustrative example, we synthesize the regression objective (8), which we will shortly show falls under the framework. The feature matrix $X \in \mathbb{R}^{128 \times 20}$ and ground-truth parameters $M^\star \in \mathbb{R}^{20 \times 10}, w^\star \in \mathbb{R}^{10}$ are randomly generated i.i.d. from the standard normal distribution, and the label is created as $y_i = \phi(x_i^\top M^\star)w^\star + \epsilon_i$ with $\epsilon_i \sim \mathcal{N}(0, 1)$. We set the regularization weight to be $0.1$.

To construct the visualization, we follow the methodology introduced in Chen et al. (2023). At selected training iteration $k$, we fix the current iterate $M_k$ and consider a two-dimensional reduction of the parameter space spanned by two randomly generated distinct directions $d_1, d_2 \in \mathbb{R}^{m \times n}$. We then sweep the scalar coefficients $(\eta_1, \eta_2)$ over a uniform grid and evaluate the objective at points of the form $M_k + \eta_1 d_1 + \eta_2 d_2$, while holding the second-layer weights fixed at either the current iterate $w_k$ for the joint objective $f(\cdot, w_k)$, or at the best-response $w^\star(M)$ for the Stackelberg objective $\Phi$. This produces a local, lower-dimensional op-

Landscapes of Joint and Stackelberg Objectives

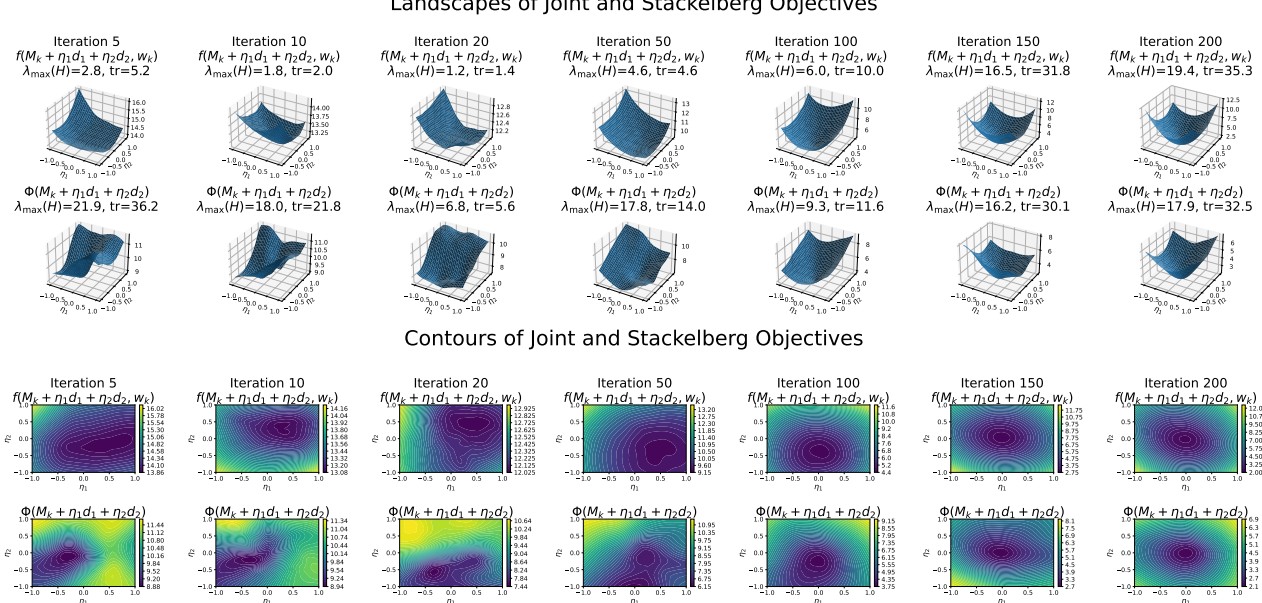

Contours of Joint and Stackelberg Objectives

*Figure 1.* Optimization landscape and contour. The Stackelberg objective (bottom row) consistently shows a strong gradient direction away from the origin and a sharper curvature (measured by the largest eigenvalue and trace of the Hessian) when the iterates are far from convergence (iterations 100 and earlier according to Figure 2).

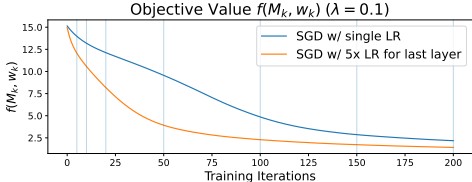

*Figure 2.* Objective function convergence under single learning rate and non-uniform learning rates, using the same training trajectories as those for generating Figure 1. Non-uniform learning rates with the final layer updating 5 times faster shows improved convergence. (The optimizer here is standard SGD for direct alignment with our theory. In the later experiments, we adopt RMSProp to demonstrate the compatibility of our theoretical insight with advanced optimizers commonly used in practice.)

timization landscape (Figure 1) that captures the geometry encountered by the body layer in the training process.

Figure 1 reveals a pronounced difference between the geometry of the joint objective and that of the Stackelberg objective, particularly during the early stages of training. When $M_k$ is far from convergence (e.g., during iterations 0–100 when the loss is still high according to Figure 2), the objective $f(\cdot, w_k)$ exhibits a notably flat landscape around the current iterate, with a minimizer close to the origin. This implies that the gradient $\nabla_M f(M_k, w_k)$ is misleadingly small without a clear informative direction. The small gradient magnitude, along with weak local curvature quantified by the largest eigenvalue and trace of the Hessian, can

impede the learning of the body-layer parameters.

In contrast, the Stackelberg objective $\Phi$ exhibits significantly sharper curvature. Especially near the origin, $\Phi$ displays a clear descent direction with larger curvature, resulting in stronger gradient signals for the body-layer updates. This allows for a faster escape from spurious flat regions and accelerates convergence during the transient phase of training. As training progresses and the body (approximately) converges, the landscapes of $f(\cdot, w_k)$ and $\Phi$ become more aligned.

# 5. Applications and Experiments

We discuss a range of problems in supervised learning and reinforcement learning that take the form of (1) and satisfy Assumptions 3.1-3.5, enabling the application of the convergence results established in Section 3. We also present experimental results illustrating the effect of non-uniform learning rates in these applications.

## 5.1. Regression

Suppose that we have a set of $N$ feature vectors $\{x_i \in \mathbb{R}^m\}$ and their ground-truth labels $\{y_i \in \mathbb{R}\}$, represented in a matrix form as $X = [x_1^\top; \cdots; x_N^\top] \in \mathbb{R}^{N \times m}$ and $Y = [y_1; \cdots; y_N] \in \mathbb{R}^N$. While our analysis generally holds for neural networks of any depth, for simplicity of exposition, we focus on the two-layer case, where $M \in \mathcal{M} \subseteq \mathbb{R}^{m \times n}$ denotes the weight matrix of the first layer and $w \in \mathcal{W} \subseteq$

$\mathbb{R}^n$ the weight of the second layer. When the input to the neural network is $x \in \mathbb{R}^m$, the output is $\phi(x^\top M)w$. The regression objective for fitting $M, w$, measured by squared $\ell_2$ norm, is

$$\min_{M,w} f(M,w) \triangleq \|\phi(XM)w - Y\|_2^2 + \frac{\lambda}{2}\|w\|_2^2. \quad (8)$$

We add a small regularization on $\|w\|^2$ to ensure $\lambda$-strong convexity with respect to $w$. Note that if the feature and first layer's weight are such that $\phi(XM)$ is over-determined, i.e. $\phi(XM)^\top \phi(XM)$ is invertible, then the unregularized problem itself is already strongly convex (with respect to $w$). In this case, the strong convexity constant is given by the smallest eigenvalue of $\phi(XM)^\top \phi(XM)$ and we can safely remove the regularization.

Let $\phi'(XM)$ denote a (Clarke) subgradient of the activation function $\phi$ at $XM$. The (sub)gradients of the objective are

$$\nabla_M^{\text{sub}} f(M,w) = 2X^\top\big(((\phi(XM)w - Y)w^\top) \odot \phi'(XM)\big),$$
$$\nabla_w f(M,w) = 2\phi(XM)^\top \phi(XM)w - 2\phi(XM)^\top Y,$$

where $\odot$ denotes element-wise multiplication.

In practice, we often compute stochastic gradient estimates by sampling a mini-batch $(x_i, y_i)$, e.g. $\mathcal{B} \subset \{1, \ldots, N\}$, and then forming the partial gradients

$$G_M(M, w, \mathcal{B}) = \frac{2N}{|\mathcal{B}|} X_{\mathcal{B}}^\top \cdot$$
$$\Big(((\phi(X_{\mathcal{B}}M)w - Y_{\mathcal{B}})w^\top) \odot \phi'(X_{\mathcal{B}}M)\Big),$$
$$\nabla_w \ell(M, w, \mathcal{B}) = \frac{2N}{|\mathcal{B}|} \phi(X_{\mathcal{B}}M)^\top (\phi(X_{\mathcal{B}}M)w - Y_{\mathcal{B}}),$$

where $X_{\mathcal{B}}$ and $Y_{\mathcal{B}}$ are the rows of $X$ and $Y$ corresponding to indices in $\mathcal{B}$.

By construction, these stochastic gradient operators are unbiased estimators of the full (sub)gradients

$$\mathbb{E}_{\mathcal{B}}[G_M(M, w, \mathcal{B})] = \nabla_M^{\text{sub}} f(M, w), \quad (9)$$
$$\mathbb{E}_{\mathcal{B}}[\nabla_w \ell(M, w, \mathcal{B})] = \nabla_w f(M, w), \quad (10)$$

where the expectation is taken over the random choice of the mini-batch $\mathcal{B}$. This holds because each data point is sampled uniformly, so the mini-batch gradient is simply an average over independent unbiased contributions from the individual samples. The regression problem falls under our framework, and the following lemma verifies the satisfaction of the assumptions.

**Lemma 5.1.** *Suppose that the feature vectors $\{x_i\}$ are bounded and that the activation function $\phi$ is Lipschitz continuous and satisfies that the function $a \mapsto \phi^2(a)$ is smooth. (Note that this may include activation functions that are themselves smooth, as well as those that are piecewise linear, such as ReLU and LeakyReLU.) Then, the regression problem satisfies Assumptions 3.1–3.5.*

## 5.2. Classification

We consider a standard binary classification problem with a dataset of $N$ labeled examples $\{(x_i, y_i)\}_{i=1}^N$, where $x_i \in \mathbb{R}^m$ denotes the feature vector and $y_i \in \{0, 1\}$ is the label. While again the following argument holds for neural networks of any depth, we adopt a two-layer neural network parameterization to simplify the presentation, where $M \in \mathbb{R}^{m \times n}$ and $w \in \mathbb{R}^n$ denote the weights of the two layers, respectively. Denote $o_i(M, w) = s(\phi(x_i^\top M)w)$, where $s : \mathbb{R} \to (0, 1)$ is the sigmoid function $s(t) = (1 + e^{-t})^{-1}$. We consider the logistic loss below

$$\min_{M,w} f(M,w) \triangleq \frac{1}{N}\sum_{i=1}^N \Big( -y_i \log o_i(M, w) \quad (11)$$
$$- (1 - y_i) \log\big(1 - o_i(M, w)\big)\Big) + \frac{\lambda}{2}\|w\|_2^2,$$

again with a small regularization on the last layer.

The (sub)gradients of the objective are

$$\nabla_M^{\text{sub}} f(M,w) = \frac{1}{N} X^\top\Big((s(\phi(XM)w) - y)w^\top \odot \phi'(XM)\Big),$$
$$\nabla_w f(M,w) = \frac{1}{N}\phi(XM)^\top\big(s(\phi(XM)w) - y\big) + \lambda w.$$

As in the regression setting, unbiased stochastic gradients can be obtained via a mini-match of randomly drawn samples. We again show that the general framework abstracts this problem exactly with all assumptions verified to hold.

**Lemma 5.2.** *Suppose that the feature vectors $\{x_i\}$ all have bounded norms, and that the activation is Lipschitz, differentiable, and smooth. Then, the classification problem satisfies Assumptions 3.1–3.5.*

## 5.3. Gradient-Based Temporal Difference Learning

We next discuss a policy evaluation method in RL, namely, temporal-difference learning with gradient correction (TDC). Consider an infinite-horizon discounted-reward Markov decision process with state space $\mathcal{S}$, action space $\mathcal{A}$, transition probability kernel $\mathcal{P} : \mathcal{S} \times \mathcal{A} \to \Delta_{\mathcal{S}}$, and reward function $r : \mathcal{S} \times \mathcal{A} \to [0, 1]$. We would like to compute the value function of a fixed policy $\pi \in \Delta_{\mathcal{A}}^{\mathcal{S}}$.

We use $d_\pi$ to denote the stationary distribution over states induced by policy $\pi$, and define the Bellman operator $T^\pi$ acting on a function $V : \mathcal{S} \to \mathbb{R}$ as

$$(T^\pi V)(s) \triangleq \mathbb{E}_{a \sim \pi(\cdot|s), \, s' \sim \mathcal{P}(\cdot|s,a)}\big[r(s,a) + \gamma V(s')\big].$$

The true value function $V^\pi$ is the unique fixed point of $T^\pi$, i.e., $V^\pi = T^\pi V^\pi$.

TDC is a well-studied algorithm for minimizing the mean-squared projected Bellman error (MSPBE) under linear function approximation. Here we extend the function approximation to a two-layer neural network, where the first-layer

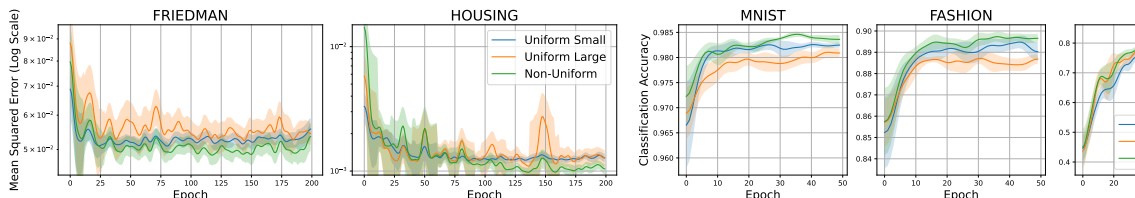

*Figure 3.* Non-uniform learning rates for regression.

*Figure 4.* Non-uniform learning rates for classification.

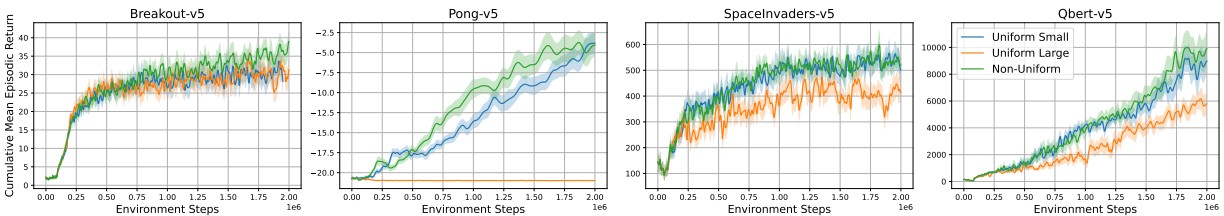

*Figure 5.* Non-uniform learning rates for policy optimization in Atari games. "Non-Uniform" indicates that we use learning rate $\alpha$ for the body and a larger rate $\beta$ for the head. "Uniform Small" and "Uniform Large" use a single learning rate equal to $\alpha$ and $\beta$, respectively.

weights are $M \in \mathbb{R}^{m \times n}$ and the final-layer weights are $w \in \mathbb{R}^n$. The feature associated with state $s$ is $\psi(s) \in \mathbb{R}^m$. From the viewpoint of the last layer, the goal is to solve a policy evaluation problem under linear function approximation, where state $s$ is associated with the representation $\psi_M(s) \triangleq \phi(M^\top \psi(s)) \in \mathbb{R}^n$. For simplicity, we assume that the activation function $\phi$ is differentiable and smooth. The value function estimated by the neural network is

$$V_{M,w}(s) = \phi(\psi(s)^\top M)w = \psi_M^\top w.$$

For a fixed $M$, let $\Pi_M$ denote the orthogonal projection onto $\Psi_M$, which is the representation space induced by $M$

$$\Psi_M \triangleq \begin{bmatrix} - & \psi_M(s_1)^\top & - \\ & \vdots & \\ - & \psi_M(s_{|\mathcal{S}|})^\top & - \end{bmatrix} \in \mathbb{R}^{|\mathcal{S}| \times n}.$$

The MSPBE objective is

$$\min_{M,w} f(M,w) \triangleq \| \Psi_M w - \Pi_M T^\pi \Psi_M w \|_{d_\pi}^2, \quad (12)$$

where the norm $\|v\|_{d_\pi}^2 = v^\top \operatorname{diag}(d_\pi) v$ is weighted by the stationary distribution.

With the exact gradient expressions $\nabla_M f(M,w)$, $\nabla_w f(M,w)$ and their derivation deferred to Appendix E, we point out that we can obtain unbiased stochastic estimates of the gradients if we know an auxiliary variable $\mu(M,w)$ that satisfies the following equation

$$\mathbb{E}_\pi[\psi_M(s)\psi_M(s)^\top]\mu(M,w) \quad (13)$$
$$= \mathbb{E}_\pi\left[ \left( r(s,a) + \gamma \psi_M(s')^\top w - \psi_M(s)^\top w \right) \psi_M(s) \right].$$

The gradient $\nabla_M f(M,w)$ is Lipschitz under bounded $\mathcal{M}, \mathcal{W}$, implying that $\Phi$ is smooth and hence weakly convex. The lower-level strong convexity condition also holds under mild non-singularity conditions (Xu et al., 2019). Assumption 3.5 can also be verified to hold in this context. The gap between TDC and our general framework is that, since $\mu(M,w)$ is not immediately available, we do not have i.i.d. unbiased stochastic gradient estimates of $\nabla_M f(M,w)$ and $\nabla_w f(M,w)$. However, $\mu(M,w)$ can be estimated by stochastic approximation on the same time scale as $w$, which leads to an algorithm that updates the estimates of $M, w, \mu$ in a single loop[1]. Under the assumption that $\mathbb{E}_\pi[\psi_M(s)\psi_M(s)^\top]$ has uniformly lower bounded eigenvalues (by a positive constant), an analysis similar to that of Theorem 3.6 can show that the resulting algorithm (with an additional $\mu$ update) converges to a stationary point of $\Phi$, still with rate $\tilde{\mathcal{O}}(k^{-2/5})$. To our knowledge, prior work has not provided closed-form gradient expressions for TDC under neural network function approximation, nor an accompanying convergence analysis. Our work fills this gap.

### 5.4. Experimental Results

We evaluate the effect of non-uniform learning rates empirically and present the results in Figures 3-5. Our first two sets of experiments are run on regression and classification problems, where our theoretical results readily apply. The benchmarks include (i) Friedman (Friedman, 1991) and Boston housing (Harrison & Rubinfeld, 1978) datasets for regression, and (ii) MNIST, MNIST Fashion, and CIFAR10 datasets for classification. The function approximation in

---

[1]The algorithm is presented in Appendix E.

the regression experiments is a two-layer neural network with fully connected layers and ReLU activation, while in the classification experiments we use a three-layer fully-connected neural network with ReLU after the body layers and softmax after the final layer.

Note that standard temporal-difference learning and Q-learning fall outside the scope of our analysis, as their objectives cannot be expressed as the minimization of a well-defined loss function. These methods instead solve for the fixed point of an operator which is not a gradient map. Consequently, they require different analytical tools, especially when the operator to solve is not strongly monotone, as is the case under neural network function approximation. However, the limitation is theoretical rather than empirical. In Figure 5, we conduct experiments with deep Q-learning on four environments from the Atari suite, namely, Breakout, Pong, Space Invaders, and Q*bert, even though our current theory does not extend to this setting. The function approximation in the Q-learning experiments is a three-layer neural network with a 2D convolutional layer followed by two fully connected layers. The activation function is ReLU.

**Experimental Setup.** While neural network parameters are updated with SGD under diminishing step sizes in the analysis, our empirical studies use RMSProp under constant step sizes, to be in line with common practice in deep learning. We assign a small learning rate $\alpha$ to the optimizer of the body layers, and $\beta$ to the final layer, where $\alpha$ is between 1e-4 and 5e-4 (depending on the task) and $\beta$ is 3-10 times larger. All experiments compare non-uniform learning rates against two baselines: (i) all layers use a uniform learning rate equal to $\alpha$, and (ii) all layers use a uniform learning rate equal to $\beta$. This comparison isolates the effect of learning-rate asymmetry from that of overall learning-rate scaling. The three methods differ only in their learning rates and are otherwise identical, including in network architecture, initialization, and random seeds.

The figures show that non-uniform learning rates consistently perform better than or on par with a uniform learning rate: having all layers learn with rate $\alpha$ is too conservative, while scaling the learning rates of all layers up to $\beta$ leads to unstable optimization or degraded final performance.

## 6. Concluding Remarks & Future Directions

This work discusses the mechanisms by which non-uniform learning rates provide a structural advantage in specific problem instances. While our findings offer a principled justification for two-time-scale training, we do not suggest that this approach universally outperforms uniform learning rates. Relative performance remains fundamentally dependent on the underlying problem structure, and uniform rates may be comparable or more effective in many cases. Our aim is not to offer a blanket prescription, but to identify where and why non-uniform rates yield superior learning dynamics.

A promising future direction is to investigate the intersection of our Stackelberg framework with the recent innovation on the design of noise-adaptive step sizes (You et al., 2017; 2020; Hao et al., 2025) in neural network training. Specifically, exploring how incorporating noise-adaptive, layer-wise rates informs and intersects with the multi-time-scale analysis may unify structural and geometric perspectives on neural network optimization.

## Disclaimer

This paper was prepared for informational purposes by the Artificial Intelligence Research group of JPMorgan Chase & Co. and its affiliates ("JP Morgan") and is not a product of the Research Department of JP Morgan. JP Morgan makes no representation and warranty whatsoever and disclaims all liability, for the completeness, accuracy or reliability of the information contained herein. This document is not intended as investment research or investment advice, or a recommendation, offer or solicitation for the purchase or sale of any security, financial instrument, financial product or service, or to be used in any way for evaluating the merits of participating in any transaction, and shall not constitute a solicitation under any jurisdiction or to any person, if such solicitation under such jurisdiction or to such person would be unlawful.

## Impact Statement

This work provides a foundational mathematical framework for understanding the role of non-uniform learning rates in neural network optimization. By characterizing these dynamics through the lens of Stackelberg optimization and two-time-scale stochastic approximation, we contribute toward a more principled approach to architecture training. As the scope of this research is primarily theoretical and does not involve new datasets or software, we do not foresee any direct negative societal consequences.

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

# Supplementary Material
# Rethinking Neural Network Learning Rates: A Stackelberg Perspective

## Contents

# A. Frequently Used Notations and Intermediate Results

We denote the Clarke directional derivative of function $\Phi$ along the direction $D$ as

$$\Phi^\circ(M; D) = \limsup_{M' \to M, t \downarrow 0} \frac{\Phi(M' + tD) - \Phi(M')}{t}. \tag{14}$$

Similarly, we denote

$$f^\circ(M, w; D) = \limsup_{M' \to M, t \downarrow 0} \frac{f(M' + tD, w) - f(M', w)}{t}. \tag{15}$$

The Clarke subdifferential is given as

$$\partial \Phi(M) \triangleq \left\{ \nu \in \mathbb{R}^{m \times n} : \langle \nu, D \rangle \leq \Phi^\circ(M; D), \forall D \right\},$$
$$\partial_M f(M, w) \triangleq \left\{ \nu \in \mathbb{R}^{m \times n} : \langle \nu, D \rangle \leq f^\circ(M, w; D), \forall D \right\}. \tag{16}$$

We denote $\hat{M} = \operatorname{argmin}_{M'} \left\{ \Phi(M') + \mathbf{1}_{\mathcal{M}}(M') + \frac{\hat{\rho}}{2} \|M - M'\|^2 \right\}$. The Moreau envelope satisfies the following properties (Davis & Drusvyatskiy, 2019).

- Let $N_{\mathcal{M}}(\hat{M})$ denote the normal cone of $\mathcal{M}$ at $\hat{M}$. We have

$$\nabla \Phi_{1/\hat{\rho}}(M) = \hat{\rho}(M - \hat{M}), \tag{17}$$

$$\hat{\rho}(M - \operatorname{prox}_{\Phi/\hat{\rho}}(M)) \in \partial \Phi(\hat{M}) + N_{\mathcal{M}}(\hat{M}). \tag{18}$$

- Lipschitz gradient

$$\|\nabla \Phi_{1/\hat{\rho}}(M) - \nabla \Phi_{1/\hat{\rho}}(M')\| \leq \hat{\rho} \|M - M'\|$$

- For any $M \in \mathbb{R}^{m \times n}$ and $\mathcal{M} \subset \mathbb{R}^{m \times n}$, define $\operatorname{dist}(M, \mathcal{M}) \triangleq \min_{M' \in \mathcal{M}} \|M - M'\|$. We have for any $M$

$$\operatorname{dist}(0, \partial \Phi(\hat{M}) + N_{\mathcal{M}}(\hat{M})) \leq \|\nabla \Phi_{1/\rho}(M)\|. \tag{19}$$

- $\Phi_{1/\hat{\rho}}$ has $\hat{\rho}$-Lipschitz gradients (Davis & Drusvyatskiy, 2019)[Lemma 2.2]

**Lemma A.1.** *Under Assumptions 3.1 and 3.5, we have for any $M, M' \in \mathbb{R}^{m \times n}$ and $\lambda > 0$*

$$\|w^\star(M') - w^\star(M)\| \leq \frac{L}{\lambda} \|M - M'\|.$$

**Lemma A.2.** *Under Assumptions 3.1 and 3.5, the function $\Phi$ is $L_\Phi$-Lipschitz, with $L_\Phi = \frac{L(\lambda+1)}{\lambda}$, implying that for all $M$ and $\nu \in \partial \Phi(M)$*

$$\|\nu\| \leq L_\Phi.$$

# B. Proof of Theorems

## B.1. Proof of Theorem 3.6

Our analysis is built on the following intermediate results on the convergence of the body-layer and last-layer weights individually. We state the propositions here and defer their proofs to Appendix C.

**Proposition B.1** (Body-Layer Weights Convergence). *Under Assumptions 3.1-3.5, we have for all $k \geq 0$*

$$\mathbb{E}[\Phi_{1/\hat{\rho}}(M_{k+1})] \leq \mathbb{E}[\Phi_{1/\hat{\rho}}(M_k)] - \frac{(4\hat{\rho} - 5\rho)\alpha_k}{4\hat{\rho}} \mathbb{E}[\|\nabla \Phi_{1/\hat{\rho}}(M_k)\|^2]$$
$$+ \frac{L^2 \hat{\rho} \alpha_k}{\rho} \mathbb{E}[\|w_k - w^\star(M_k)\|^2] + \frac{(L_\Phi^2 + \sigma^2)\hat{\rho}\alpha_k^2}{2}.$$

**Proposition B.2** (Last-Layer Weights Convergence). *Under Assumptions 3.1-3.5 and the step size condition $\beta_k \leq \min\{\frac{\lambda}{2L^2}, \frac{2}{\lambda}\}$, we have for all $k \geq 0$*

$$\mathbb{E}[\|w_{k+1} - w^\star(M_{k+1})\|^2] \leq (1 - \lambda\beta_k)\mathbb{E}[\|w_k - w^\star(M_k)\|^2] + \frac{4L^2(L_\Phi^2 + \sigma^2)\alpha_k^2}{\lambda^3\beta_k} + 2\sigma^2\beta_k^2.$$

Adding together the bounds from Propositions B.1 and B.2, with the terms in Proposition B.2 scaled by $\frac{4L^2\hat{\rho}^2\alpha_k}{(4\hat{\rho}-5\rho)\rho\lambda\beta_k}$, we have

$$\alpha_k\mathbb{E}[\|\nabla\Phi_{1/\hat{\rho}}(M_k)\|^2]$$

$$\leq \frac{4\hat{\rho}}{(4\hat{\rho}-5\rho)}\mathbb{E}[\Phi_{1/\hat{\rho}}(M_k) - \Phi_{1/\hat{\rho}}(M_{k+1})] + \frac{4L^2\hat{\rho}^2\alpha_k}{(4\hat{\rho}-5\rho)\rho}\mathbb{E}[\|w_k - w^\star(M_k)\|^2] + \frac{2(L_\Phi^2 + \sigma^2)\hat{\rho}^2\alpha_k^2}{4\hat{\rho}-5\rho}$$

$$- \frac{4L^2\hat{\rho}^2\alpha_k}{(4\hat{\rho}-5\rho)\rho\lambda\beta_k}\mathbb{E}[\|w_{k+1} - w^\star(M_{k+1})\|^2] + \frac{4L^2\hat{\rho}^2\alpha_k}{(4\hat{\rho}-5\rho)\rho\lambda\beta_k}(1 - \lambda\beta_k)\mathbb{E}[\|w_k - w^\star(M_k)\|^2]$$

$$+ \frac{4L^2\hat{\rho}^2\alpha_k}{(4\hat{\rho}-5\rho)\rho\lambda\beta_k} \cdot \frac{4L^2(L_\Phi^2 + \sigma^2)\alpha_k^2}{\lambda^3\beta_k} + \frac{4L^2\hat{\rho}^2\alpha_k}{(4\hat{\rho}-5\rho)\rho\lambda\beta_k} \cdot 2\sigma^2\beta_k^2$$

$$= \frac{4\hat{\rho}}{(4\hat{\rho}-5\rho)}\mathbb{E}[\Phi_{1/\hat{\rho}}(M_k) - \Phi_{1/\hat{\rho}}(M_{k+1})]$$

$$+ \frac{4L^2\hat{\rho}^2\alpha_k}{(4\hat{\rho}-5\rho)\rho\lambda\beta_k}\mathbb{E}[\|w_k - w^\star(M_k)\|^2 - \|w_{k+1} - w^\star(M_{k+1})\|^2]$$

$$+ \frac{2(L_\Phi^2 + \sigma^2)\hat{\rho}^2\alpha_k^2}{4\hat{\rho}-5\rho} + \frac{16L^4(L_\Phi^2 + \sigma^2)\hat{\rho}^2\alpha_k^3}{(4\hat{\rho}-5\rho)\rho\lambda^4\beta_k^2} + \frac{16L^2\sigma^2\hat{\rho}^2\alpha_k\beta_k}{(4\hat{\rho}-5\rho)\rho\lambda}.$$

We plug in the step size conditions

$$\alpha_k = \frac{\alpha_0}{(k+1)^{3/5}}, \quad \beta_k = \frac{\beta_0}{(k+1)^{2/5}}$$

and further simplify

$$\frac{1}{(k+1)^{3/5}}\mathbb{E}[\|\nabla\Phi_{1/\hat{\rho}}(M_k)\|^2]$$

$$\leq \frac{4\hat{\rho}}{(4\hat{\rho}-5\rho)\alpha_0}\mathbb{E}[\Phi_{1/\hat{\rho}}(M_k) - \Phi_{1/\hat{\rho}}(M_{k+1})]$$

$$+ \frac{4L^2\hat{\rho}^2}{(4\hat{\rho}-5\rho)\rho\lambda\beta_0(k+1)^{1/5}}\mathbb{E}[\|w_k - w^\star(M_k)\|^2 - \|w_{k+1} - w^\star(M_{k+1})\|^2]$$

$$+ \frac{2(L_\Phi^2 + \sigma^2)\hat{\rho}^2\alpha_0}{(4\hat{\rho}-5\rho)(k+1)^{6/5}} + \frac{16L^4(L_\Phi^2 + \sigma^2)\hat{\rho}^2\alpha_0^2}{(4\hat{\rho}-5\rho)\rho\lambda^4\beta_0^2(k+1)} + \frac{16L^2\sigma^2\hat{\rho}^2\beta_0}{(4\hat{\rho}-5\rho)\rho\lambda(k+1)}.$$

Note that $\|\nabla\Phi_{1/\hat{\rho}}(M_k)\|$ can be lower bounded by $\text{dist}(0, \partial\Phi(\hat{M}_k) + N_\mathcal{M}(\hat{M}))$ as shown in (19). Summing up over

iterations, we have

$$\sum_{t=0}^{k-1} \frac{1}{(t+1)^{3/5}} \mathbb{E}\left[\left(\text{dist}\left(0, \partial\Phi(\hat{M}_t) + N_{\mathcal{M}}(\hat{M}_t)\right)\right)^2\right]$$

$$\leq \frac{4\hat{\rho}}{(4\hat{\rho} - 5\rho)\alpha_0} \sum_{t=0}^{k-1} \mathbb{E}[\Phi_{1/\hat{\rho}}(M_t) - \Phi_{1/\hat{\rho}}(M_{t+1})]$$

$$+ \sum_{t=0}^{k-1} \frac{4L^2\hat{\rho}^2}{(4\hat{\rho} - 5\rho)\rho\lambda\beta_0(t+1)^{1/5}} \mathbb{E}[\|w_t - w^\star(M_t)\|^2 - \|w_{t+1} - w^\star(M_{t+1})\|^2]$$

$$+ \left(\frac{2(L_\Phi^2 + \sigma^2)\hat{\rho}^2\alpha_0}{4\hat{\rho} - 5\rho} + \frac{16L^4(L_\Phi^2 + \sigma^2)\hat{\rho}^2\alpha_0^2}{(4\hat{\rho} - 5\rho)\rho\lambda^4\beta_0^2} + \frac{16L^2\sigma^2\hat{\rho}^2\beta_0}{(4\hat{\rho} - 5\rho)\rho\lambda}\right) \sum_{t=0}^{k-1} \frac{1}{t+1}$$

$$\leq \frac{4\hat{\rho}}{(4\hat{\rho} - 5\rho)\alpha_0} \mathbb{E}[\Phi_{1/\hat{\rho}}(M_0) - \Phi_{1/\hat{\rho}}(M_k)] + \frac{4L^2\hat{\rho}^2}{(4\hat{\rho} - 5\rho)\rho\lambda\beta_0}\|w_0 - w^\star(M_0)\|^2$$

$$+ \left(\frac{2(L_\Phi^2 + \sigma^2)\hat{\rho}^2\alpha_0}{4\hat{\rho} - 5\rho} + \frac{16L^4(L_\Phi^2 + \sigma^2)\hat{\rho}^2\alpha_0^2}{(4\hat{\rho} - 5\rho)\rho\lambda^4\beta_0^2} + \frac{16L^2\sigma^2\hat{\rho}^2\beta_0}{(4\hat{\rho} - 5\rho)\rho\lambda}\right) \sum_{t=0}^{k-1} \frac{1}{t+1}$$

$$\leq \frac{4\hat{\rho}}{(4\hat{\rho} - 5\rho)\alpha_0} \left(\Phi_{1/\hat{\rho}}(M_0) - \Phi(M^\star)\right) + \frac{4L^2\hat{\rho}^2}{(4\hat{\rho} - 5\rho)\rho\lambda\beta_0}\|w_0 - w^\star(M_0)\|^2$$

$$+ \left(\frac{2(L_\Phi^2 + \sigma^2)\hat{\rho}^2\alpha_0}{4\hat{\rho} - 5\rho} + \frac{16L^4(L_\Phi^2 + \sigma^2)\hat{\rho}^2\alpha_0^2}{(4\hat{\rho} - 5\rho)\rho\lambda^4\beta_0^2} + \frac{16L^2\sigma^2\hat{\rho}^2\beta_0}{(4\hat{\rho} - 5\rho)\rho\lambda}\right) \sum_{t=0}^{k-1} \frac{1}{t+1}, \tag{20}$$

where the second inequality follows from the fact that $\frac{4L^2\hat{\rho}^2}{(4\hat{\rho}-5\rho)\rho\lambda\beta_0(t+1)^{1/3}}$ is a decaying sequence and $\|w_t - w^\star(M_t)\|^2 \geq 0$ for all $t$, and the third inequality is due to the simple identity

$$\Phi_{1/\hat{\rho}}(M_k) = \min_M \{\Phi(M) + \frac{\hat{\rho}}{2}\|M - M_k\|^2\} \geq \min_M \Phi(M) + \min_M \|M - M_k\|^2 \geq \Phi(M^\star).$$

The following bounds on summations of decaying sequences are standard results and easily verifiable

$$\sum_{t=0}^{k-1} \frac{1}{(t+1)^{3/5}} \geq \frac{(k+1)^{2/5}}{2}, \quad \sum_{t=0}^{k-1} \frac{1}{t+1} \leq 2\log(k+1). \tag{21}$$

Combining (20) and (21),

$$\min_{t<k} \mathbb{E}\left[\left(\text{dist}\left(0, \partial\Phi(\hat{M}_t) + N_{\mathcal{M}}(\hat{M}_t)\right)\right)^2\right]$$

$$\leq \frac{1}{\sum_{t'=0}^{k-1} \frac{1}{(t'+1)^{3/5}}} \sum_{t=0}^{k-1} \frac{1}{(t+1)^{3/5}} \mathbb{E}[(\text{dist}(0, \partial\Phi(\hat{M}_t) + +N_{\mathcal{M}}(\hat{M}_t)))^2]$$

$$\leq \frac{1}{\sum_{t'=0}^{k-1} \frac{1}{(t'+1)^{3/5}}} \left(\frac{4\hat{\rho}}{(4\hat{\rho} - 5\rho)\alpha_0}\left(\Phi_{1/\hat{\rho}}(M_0) - \Phi(M^\star)\right) + \frac{4L^2\hat{\rho}^2}{(4\hat{\rho} - 5\rho)\rho\lambda\beta_0}\|w_0 - w^\star(M_0)\|^2\right)$$

$$+ \frac{\sum_{t'=0}^{k-1} \frac{1}{t'+1}}{\sum_{t'=0}^{k-1} \frac{1}{(t'+1)^{3/5}}} \left(\frac{2(L_\Phi^2 + \sigma^2)\hat{\rho}^2\alpha_0}{4\hat{\rho} - 5\rho} + \frac{16L^4(L_\Phi^2 + \sigma^2)\hat{\rho}^2\alpha_0^2}{(4\hat{\rho} - 5\rho)\rho\lambda^4\beta_0^2} + \frac{16L^2\sigma^2\hat{\rho}^2\beta_0}{(4\hat{\rho} - 5\rho)\rho\lambda}\right)$$

$$\leq \left(\frac{8\hat{\rho}}{(4\hat{\rho} - 5\rho)\alpha_0}\left(\Phi_{1/\hat{\rho}}(M_0) - \Phi(M^\star)\right) + \frac{8L^2\hat{\rho}^2}{(4\hat{\rho} - 5\rho)\rho\lambda\beta_0}\|w_0 - w^\star(M_0)\|^2\right) \frac{1}{(k+1)^{2/5}}$$

$$+ \left(\frac{8(L_\Phi^2 + \sigma^2)\hat{\rho}^2\alpha_0}{4\hat{\rho} - 5\rho} + \frac{64L^4(L_\Phi^2 + \sigma^2)\hat{\rho}^2\alpha_0^2}{(4\hat{\rho} - 5\rho)\rho\lambda^4\beta_0^2} + \frac{64L^2\sigma^2\hat{\rho}^2\beta_0}{(4\hat{\rho} - 5\rho)\rho\lambda}\right) \frac{\log(k+1)}{(k+1)^{2/5}},$$

where in the last inequality we plug in the step size condition $\alpha_0 \leq \beta_0 \leq 1$.

$\square$

## B.2. Proof of Theorem 3.7

The per-iteration convergence of the last-layer weights is analyzed in Proposition B.2. Below we establish the convergence of the body-layer weights. The proof of Theorem 3.7 combines the results of the propositions and bound the joint decay of the body- and last-layer residuals through a coupled two-time-scale lemma (Lemma B.4).

**Proposition B.3** (Body-Layer Weights Convergence). *Suppose that the step size $\alpha_k$ satisfies $\alpha_k \leq 1$ for all $k$. Under Assumptions 3.1-3.5 and the $\lambda_\Phi$-strong convexity of $\Phi$, we have for all $k \geq 0$*

$$
\begin{aligned}
\mathbb{E}[\|M_{k+1} - M^\star\|^2] \leq {} & (1 - \frac{\lambda_\Phi \alpha_k}{4})\mathbb{E}[\|M_k - M^\star\|^2] \\
& + \frac{(4 + \lambda_\Phi)L^2 \alpha_k}{\lambda_\Phi}\mathbb{E}[\|w_k - w^\star(M_k)\|^2] + \frac{(4 + \lambda_\Phi)(L_\Phi^2 + \sigma^2)\alpha_k^2}{4}.
\end{aligned}
$$

**Lemma B.4** (Adapted from Lemma 4 of Zeng et al. (2024)). *Let $\{a_k, b_k, c_k, d_k, e_k\}$ be non-negative sequences satisfying $\frac{a_{k+1}}{d_{k+1}} \leq \frac{a_k}{d_k} < 1$, for all $k \geq 0$. Let $\{x_k\}, \{y_k\}$ be two non-negative sequences. Suppose that $x_k, y_k$ satisfy the following coupled inequalities*

$$
x_{k+1} \leq (1 - a_k)x_k + b_k y_k + c_k, \quad y_{k+1} \leq (1 - d_k)y_k + e_k. \tag{22}
$$

*In addition, assume that there exists a constant $A > 0$ such that*

$$
Aa_k - b_k - \frac{Aa_k^2}{d_k} \geq 0, \quad \text{for all } k \geq 0. \tag{23}
$$

*Then we have for all $k \geq 0$*

$$
x_k \leq \left(x_0 + \frac{Aa_0}{d_0}y_0\right)\prod_{t=0}^{k-1}(1 - \frac{a_t}{2}) + \sum_{\ell=0}^{k-1}\left(c_\ell + \frac{Aa_\ell e_\ell}{d_\ell}\right)\prod_{t=\ell+1}^{k-1}(1 - \frac{a_t}{2}).
$$

To put the coupled per-iteration convergence bounds from Propositions B.3 and B.2 into the framework of Lemma B.4, we can set

$$
\begin{aligned}
x_k &= \mathbb{E}\left[\|M_k - M^\star\|^2\right], \\
y_k &= \mathbb{E}\left[\|w_k - w^\star(M_k)\|^2\right].
\end{aligned}
$$

The coefficients in Lemma B.4 can be identified as

$$
\begin{aligned}
a_k &= \frac{\lambda_\Phi \alpha_k}{4}, \\
b_k &= \frac{(4 + \lambda_\Phi)L^2}{\lambda_\Phi}\alpha_k, \\
c_k &= \frac{(4 + \lambda_\Phi)(L_\Phi^2 + \sigma^2)}{4}\alpha_k^2, \\
d_k &= \lambda\beta_k, \\
e_k &= \frac{4L^2(L_\Phi^2 + \sigma^2)\alpha_k^2}{\lambda^3 \beta_k} + 2\sigma^2 \beta_k^2.
\end{aligned}
$$

Under the step size condition $\frac{\alpha_0}{\beta_0} \leq \frac{2\lambda}{\lambda_\Phi}$, we have $\frac{a_k^2}{d_k} \leq \frac{\alpha_k}{2}$, so we can set $A = \frac{2b_k}{a_k} = \frac{8(4 + \lambda_\Phi)L^2}{\lambda_\Phi^2}$ to satisfy (23). This allows

us to apply Lemma B.4 and conclude

$$\mathbb{E}[\|M_k - M^\star\|^2]$$

$$\leq \left( \mathbb{E}[\|M_0 - M^\star\|^2] + \frac{8(4 + \lambda_\Phi)L^2}{\lambda_\Phi^2} \cdot \frac{\lambda_\Phi \alpha_0}{4\lambda\beta_0} \mathbb{E}[\|w_0 - w^\star(M_0)\|^2] \right) \prod_{t=0}^{k-1} \left( 1 - \frac{\lambda_\Phi \alpha_t}{8} \right)$$

$$+ \sum_{\ell=0}^{k-1} \left( \frac{(4+\lambda_\Phi)(L_\Phi^2 + \sigma^2)}{4} \alpha_\ell^2 + \frac{8(4+\lambda_\Phi)L^2}{\lambda_\Phi^2} \cdot \frac{\lambda_\Phi \alpha_\ell}{4\lambda\beta_\ell} \cdot e_\ell \right) \prod_{t=\ell+1}^{k-1} \left( 1 - \frac{\lambda_\Phi \alpha_t}{8} \right)$$

$$\leq \left( \mathbb{E}[\|M_0 - M^\star\|^2] + \frac{4(4+\lambda_\Phi)L^2}{\lambda_\Phi^2} \mathbb{E}[\|w_0 - w^\star(M_0)\|^2] \right) \prod_{t=0}^{k-1} \left( 1 - \frac{\lambda_\Phi \alpha_t}{8} \right)$$

$$+ \sum_{\ell=0}^{k-1} \left( \frac{(4+\lambda_\Phi)(L_\Phi^2 + \sigma^2)\alpha_0^2}{4(\ell+h+1)^2} + \frac{4(4+\lambda_\Phi)L^2}{\lambda_\Phi^2(\ell+h+1)^{1/3}} \cdot \frac{\frac{4L^2(L_\Phi^2+\sigma^2)\alpha_0^2}{\lambda^3\beta_0} + 2\sigma^2\beta_0^2}{(\ell+h+1)^{4/3}} \right) \prod_{t=\ell+1}^{k-1} \left( 1 - \frac{\lambda_\Phi \alpha_t}{8} \right)$$

$$= \left( \|M_0 - M^\star\|^2 + \frac{4(4+\lambda_\Phi)L^2}{\lambda_\Phi^2} \|w_0 - w^\star(M_0)\|^2 \right) \prod_{t=0}^{k-1} \left( 1 - \frac{\lambda_\Phi \alpha_t}{8} \right)$$

$$+ \sum_{\ell=0}^{k-1} \frac{C}{(\ell+h+1)^{5/3}} \prod_{t=\ell+1}^{k-1} \left( 1 - \frac{\lambda_\Phi \alpha_t}{8} \right), \tag{24}$$

where to derive the second inequality we plug in the step size and use the relationship $\alpha_k \leq \beta_k$ and $\frac{\lambda_\Phi \alpha_0}{4\lambda\beta_0} \leq 1/2$, and the equation follows from the constant definition

$$C = \frac{(4+\lambda_\Phi)(L_\Phi^2+\sigma^2)\alpha_0^2}{4} + \frac{4(4+\lambda_\Phi)L^2}{\lambda_\Phi^2} \left( \frac{4L^2(L_\Phi^2+\sigma^2)\alpha_0^2}{\lambda^3\beta_0} + 2\sigma^2\beta_0^2 \right).$$

Since $1 + c \leq \exp(c)$ for any scalar $c$, we have

$$\prod_{t=0}^{k-1} \left( 1 - \frac{\lambda_\Phi \alpha_t}{8} \right) \leq \prod_{t=0}^{k-1} \exp\left( -\frac{\lambda_\Phi \alpha_t}{8} \right)$$

$$= \exp\left( -\frac{\lambda_\Phi}{8} \sum_{t=0}^{k-1} \frac{\alpha_0}{t+h+1} \right) \leq \exp\left( \frac{\lambda_\Phi \alpha_0}{8} \log\left( \frac{k+h+1}{h+1} \right) \right)$$

$$= \left( \frac{h+1}{k+h+1} \right)^{\lambda_\Phi \alpha_0/8} \leq \frac{h+1}{k+h+1}, \tag{25}$$

where the last inequality is due to the step size condition $\alpha_0 \geq \frac{8}{\lambda_\Phi}$.

Similarly,

$$\prod_{t=\ell+1}^{k-1} (1 - \frac{\lambda_\Phi \alpha_t}{8}) \leq \frac{\ell+h+2}{k+h+1} \leq \frac{2(\ell+h+1)}{k+h+1}. \tag{26}$$

Combining (24)-(26),

$$\mathbb{E}[\|M_k - M^\star\|^2]$$

$$\leq \left( \|M_0 - M^\star\|^2 + \frac{4(4+\lambda_\Phi)L^2}{\lambda_\Phi^2} \|w_0 - w^\star(M_0)\|^2 \right) \frac{h+1}{k+h+1} + \frac{2C}{k+h+1} \sum_{\ell=0}^{k-1} \frac{1}{(\ell+h+1)^{2/3}}$$

$$\leq \left( \|M_0 - M^\star\|^2 + \frac{4(4+\lambda_\Phi)L^2}{\lambda_\Phi^2} \|w_0 - w^\star(M_0)\|^2 \right) \frac{h+1}{k+h+1} + \frac{2C}{(k+h+1)^{2/3}}, \tag{27}$$

where the second inequality is due to the relationship $\sum_{t=0}^{t'} \frac{1}{(t+1)^{2/3}} \leq \frac{(t'+1)^{1/3}}{3}$.

$\square$

# C. Proof of Propositions

## C.1. Proof of Proposition B.1

Define $\hat{M}_k \triangleq \text{prox}_{\Phi/\hat{\rho}}(M_k) = \text{argmin}_M \{\Phi(M) + \mathbf{1}_\mathcal{M}(M) + \frac{\hat{\rho}}{2}\|M_k - M\|^2\}$.

By the definition of $\Phi_{1/\hat{\rho}}$,

$$\Phi_{1/\hat{\rho}}(M_{k+1}) \leq \Phi(\hat{M}_k) + \frac{\hat{\rho}}{2}\|\hat{M}_k - M_{k+1}\|^2$$

$$= \Phi(\hat{M}_k) + \frac{\hat{\rho}}{2}\|\hat{M}_k - \text{proj}_\mathcal{M}(M_k - \alpha_k G_M(M_k, w_k, \xi_k))\|^2$$

$$\leq \Phi(\hat{M}_k) + \frac{\hat{\rho}}{2}\|\hat{M}_k - M_k + \alpha_k G_M(M_k, w_k, \xi_k)\|^2$$

$$\leq \Phi(\hat{M}_k) + \frac{\hat{\rho}}{2}\|\hat{M}_k - M_k\|^2 + \hat{\rho}\alpha_k\langle\hat{M}_k - M_k, G_M(M_k, w_k, \xi_k)\rangle + \frac{\hat{\rho}\alpha_k^2}{2}\|G_M(M_k, w_k, \xi_k)\|^2$$

$$= \Phi_{1/\hat{\rho}}(M_k) + \hat{\rho}\alpha_k\langle\hat{M}_k - M_k, G_M(M_k, w_k, \xi_k) - G_M(M_k, w^\star(M_k), \xi_k)\rangle$$

$$+ \hat{\rho}\alpha_k\langle\hat{M}_k - M_k, G_M(M_k, w^\star(M_k), \xi_k)\rangle + \frac{\hat{\rho}\alpha_k^2}{2}\|G_M(M_k, w_k, \xi_k)\|^2$$

where the second inequality is due to the fact that the projection to a closed convex set is non-expansive, and the final equation follows from the definition of $\hat{M}_k$.

Let $\nu_k \triangleq \mathbb{E}_{\xi_k \sim \mathcal{D}}[G_M(M_k, w^\star(M_k), \xi_k)]$. It is clear that $\nu_k \in \partial\Phi(M_k)$ due to Lemma 3.2, i.e. $\nu_k$ is a subgradient of $\Phi$ with respect to $M$ at $M_k$. Taking the expectation, we have from the inequality above

$$\mathbb{E}[\Phi_{1/\hat{\rho}}(M_{k+1})]$$

$$\leq \mathbb{E}[\Phi_{1/\hat{\rho}}(M_k)] + \hat{\rho}\alpha_k\mathbb{E}[\langle\hat{M}_k - M_k, G_M(M_k, w_k, \xi_k) - G_M(M_k, w^\star(M_k), \xi_k)\rangle]$$

$$+ \hat{\rho}\alpha_k\mathbb{E}[\langle\hat{M}_k - M_k, \nu_k\rangle] + \frac{\hat{\rho}\alpha_k^2}{2}\mathbb{E}[\|G_M(M_k, w_k, \xi_k)\|^2]$$

$$\leq \mathbb{E}[\Phi_{1/\hat{\rho}}(M_k)] + \frac{\hat{\rho}\rho\alpha_k}{4}\mathbb{E}[\|\hat{M}_k - M_k\|^2] + \frac{\hat{\rho}\alpha_k}{\rho}\mathbb{E}[\|G_M(M_k, w_k, \xi_k) - G_M(M_k, w^\star(M_k), \xi_k)\|^2]$$

$$+ \hat{\rho}\alpha_k\mathbb{E}[\langle\hat{M}_k - M_k, \nu_k\rangle] + \frac{\hat{\rho}\alpha_k^2}{2}\mathbb{E}[\|G_M(M_k, w_k, \xi_k)\|^2]$$

$$= \mathbb{E}[\Phi_{1/\hat{\rho}}(M_k)] + \frac{\hat{\rho}\rho\alpha_k}{4}\mathbb{E}[\|\hat{M}_k - M_k\|^2] + \frac{\hat{\rho}\alpha_k}{\rho}\mathbb{E}[\|G_M(M_k, w_k, \xi_k) - G_M(M_k, w^\star(M_k), \xi_k)\|^2]$$

$$+ \hat{\rho}\alpha_k\mathbb{E}[\langle\hat{M}_k - M_k, \nu_k\rangle] + \frac{\hat{\rho}\alpha_k^2}{2}\mathbb{E}[\|\nu_k\|^2] + \frac{\hat{\rho}\alpha_k^2}{2}\mathbb{E}[\|G_M(M_k, w_k, \xi_k) - \nu_k\|^2]$$

$$\leq \mathbb{E}[\Phi_{1/\hat{\rho}}(M_k)] + \frac{\hat{\rho}\rho\alpha_k}{4}\mathbb{E}[\|\hat{M}_k - M_k\|^2] + \frac{L^2\hat{\rho}\alpha_k}{\rho}\mathbb{E}[\|w_k - w^\star(M_k)\|^2]$$

$$+ \hat{\rho}\alpha_k\mathbb{E}[\Phi(\hat{M}_k) - \Phi(M_k) + \frac{\rho}{2}\|\hat{M}_k - M_k\|^2] + \frac{\hat{\rho}\alpha_k^2}{2}\mathbb{E}[\|\nu_k\|^2] + \frac{\hat{\rho}\alpha_k^2}{2}\mathbb{E}[\|G_M(M_k, w_k, \xi_k) - \nu_k\|^2]$$

$$\leq \mathbb{E}[\Phi_{1/\hat{\rho}}(M_k)] + \frac{\hat{\rho}\rho\alpha_k}{4}\mathbb{E}[\|\hat{M}_k - M_k\|^2] + \frac{L^2\hat{\rho}\alpha_k}{\rho}\mathbb{E}[\|w_k - w^\star(M_k)\|^2]$$

$$+ \hat{\rho}\alpha_k\mathbb{E}[\Phi(\hat{M}_k) - \Phi(M_k) + \frac{\rho}{2}\|\hat{M}_k - M_k\|^2] + \frac{(L_\Phi^2 + \sigma^2)\hat{\rho}\alpha_k^2}{2}$$

$$= \mathbb{E}[\Phi_{1/\hat{\rho}}(M_k)] + \frac{L^2\hat{\rho}\alpha_k}{\rho}\mathbb{E}[\|w_k - w^\star(M_k)\|^2]$$

$$+ \hat{\rho}\alpha_k\mathbb{E}[\Phi(\hat{M}_k) - \Phi(M_k) + \frac{3\rho}{4}\|\hat{M}_k - M_k\|^2] + \frac{(L_\Phi^2 + \sigma^2)\hat{\rho}\alpha_k^2}{2},$$

where the third inequality follows from (5) and the Lipschitz continuity of $G_M$ in Assumption 3.5, and the last inequality follows from Assumption 3.4 on bounded variance and Lemma A.2 on the magnitude of $\|\nu_k\|$.

As the function $M \mapsto \Phi(M) + \frac{\hat{\rho}}{2}\|M - M_k\|^2$ is $(\hat{\rho} - \rho)$-strongly convex and $\hat{M}_k$ is the optimizer of this function (i.e. 0 is in the subdifferential at $\hat{M}_k$), we have

$$
\begin{aligned}
\Phi(M_k) &- \Phi(\hat{M}_k) - \frac{3\rho}{4}\|\hat{M}_k - M_k\|^2 \\
&= \left(\Phi(M_k) + \frac{\hat{\rho}}{2}\|M_k - M_k\|^2\right) - \left(\Phi(\hat{M}_k) + \frac{\hat{\rho}}{2}\|\hat{M}_k - M_k\|^2\right) + \frac{2\hat{\rho} - 3\rho}{4}\|\hat{M}_k - M_k\|^2 \\
&\geq \frac{\hat{\rho} - \rho}{2}\|\hat{M}_k - M_k\|^2 + \frac{2\hat{\rho} - 3\rho}{4}\|\hat{M}_k - M_k\|^2 \\
&= \frac{4\hat{\rho} - 5\rho}{4}\|\hat{M}_k - M_k\|^2 \\
&= \frac{4\hat{\rho} - 5\rho}{4\hat{\rho}^2}\|\nabla\Phi_{1/\hat{\rho}}(M_k)\|^2.
\end{aligned}
\tag{28}
$$

where the last equation is due to (17).

Combining the two inequalities above,

$$
\begin{aligned}
\mathbb{E}[\Phi_{1/\hat{\rho}}(M_{k+1})] \leq{}& \mathbb{E}[\Phi_{1/\hat{\rho}}(M_k)] + \frac{L^2\hat{\rho}\alpha_k}{\rho}\mathbb{E}[\|w_k - w^\star(M_k)\|^2] \\
&- \frac{(4\hat{\rho} - 5\rho)\alpha_k}{4\hat{\rho}}\mathbb{E}[\|\nabla\Phi_{1/\hat{\rho}}(M_k)\|^2] + \frac{(L_\Phi^2 + \sigma^2)\hat{\rho}\alpha_k^2}{2}.
\end{aligned}
$$

$\square$

### C.2. Proof of Proposition B.2

By the update rule of $w_{k+1}$,

$$
\begin{aligned}
\|w_{k+1} &- w^\star(M_{k+1})\|^2 \\
&= \|\operatorname{proj}_{\mathcal{W}}(w_k - \beta_k\nabla_w\ell(M_k, w_k, \xi_k)) - w^\star(M_k) + (w^\star(M_k) - w^\star(M_{k+1}))\|^2 \\
&\leq \|w_k - \beta_k\nabla_w\ell(M_k, w_k, \xi_k) - w^\star(M_k) + (w^\star(M_k) - w^\star(M_{k+1}))\|^2 \\
&= \|w_k - w^\star(M_k) - \beta_k\nabla_w f(M_k, w_k) + \beta_k\big(\nabla_w f(M_k, w_k) - \nabla_w\ell(M_k, w_k, \xi_k)\big) + (w^\star(M_k) - w^\star(M_{k+1}))\|^2 \\
&\leq \|w_k - w^\star(M_k) - \beta_k\nabla_w f(M_k, w_k)\|^2 + 2\beta_k^2\|\nabla_w f(M_k, w_k) - \nabla_w\ell(M_k, w_k, \xi_k)\|^2 \\
&\quad + 2\|w^\star(M_k) - w^\star(M_{k+1})\|^2 \\
&\quad + 2\beta_k\langle w_k - w^\star(M_k) - \beta_k\nabla_w f(M_k, w_k), \nabla_w f(M_k, w_k) - \nabla_w\ell(M_k, w_k, \xi_k)\rangle \\
&\quad + 2\langle w_k - w^\star(M_k) - \beta_k\nabla_w f(M_k, w_k), w^\star(M_k) - w^\star(M_{k+1})\rangle,
\end{aligned}
\tag{29}
$$

where the first inequality is due to the fact that the projection to a closed convex set is non-expansive.

We have the following bound on the first term of (29).

$$
\begin{aligned}
\|w_k &- w^\star(M_k) - \beta_k\nabla_w f(M_k, w_k)\|^2 \\
&= \|w_k - w^\star(M_k)\|^2 + \beta_k^2\|\nabla_w f(M_k, w_k) - \nabla_w f(M_k, w^\star(M_k))\|^2 - 2\beta_k\langle w_k - w^\star(M_k), \nabla_w f(M_k, w_k)\rangle \\
&\leq \|w_k - w^\star(M_k)\|^2 + \beta_k^2\|\nabla_w f(M_k, w_k) - \nabla_w f(M_k, w^\star(M_k))\|^2 - 2\lambda\beta_k\|w_k - w^\star(M_k)\|^2 \\
&\leq (1 - 2\lambda\beta_k)\|w_k - w^\star(M_k)\|^2 + L^2\beta_k^2\|w_k - w^\star(M_k)\|^2 \\
&\leq (1 - \frac{3\lambda\beta_k}{2})\|w_k - w^\star(M_k)\|^2,
\end{aligned}
\tag{30}
$$

where the first inequality is due to the fact that $f(M, \cdot)$ is $\lambda$-strongly convex with respect to $w$ for any $M$, the second inequality is due to Assumption 3.5, and the final inequality holds under the step size condition $\beta_k \leq \frac{\lambda}{2L^2}$.

The second term is bounded of (29) in expectation due to Assumption 3.4.

$$
2\beta_k^2\mathbb{E}[\|\nabla_w f(M_k, w_k) - \nabla_w\ell(M_k, w_k, \xi_k)\|^2] \leq 2\sigma^2\beta_k^2.
\tag{31}
$$

To bound the third term of (29), we leverage the Lipschitz continuity of the mapping $w^\star$ established in Lemma A.1. Denoting $\nu_k = \mathbb{E}_{\xi_k \sim \mathcal{D}}[G_M(M_k, w_k, \xi_k)]$, we have

$$
\begin{aligned}
\mathbb{E}[\|w^\star(M_k) - w^\star(M_{k+1})\|^2] &\leq \frac{L^2}{\lambda^2} \mathbb{E}[\|M_k - M_{k+1}\|^2] \\
&= \frac{L^2 \alpha_k^2}{\lambda^2} \mathbb{E}[\|G_M(M_k, w_k, \xi_k)\|^2] \\
&= \frac{L^2 \alpha_k^2}{\lambda^2} \mathbb{E}[\|\nu_k\|^2] + \frac{L^2 \alpha_k^2}{\lambda^2} \mathbb{E}[\|G_M(M_k, w_k, \xi_k) - \nu_k\|^2] \\
&\leq \frac{L^2 (L_\Phi^2 + \sigma^2) \alpha_k^2}{\lambda^2},
\end{aligned}
\tag{32}
$$

where the last inequality follows from Assumption 3.4 and Lemma A.2.

The fourth term of (29) vanishes in expectation

$$
\begin{aligned}
&2\beta_k \mathbb{E}[\langle w_k - w^\star(M_k) - \beta_k \nabla_w f(M_k, w_k), \nabla_w f(M_k, w_k) - \nabla_w \ell(M_k, w_k, \xi_k)\rangle] \\
&= 2\beta_k \mathbb{E}[\langle w_k - w^\star(M_k) - \beta_k \nabla_w f(M_k, w_k), \nabla_w f(M_k, w_k) - \mathbb{E}[\nabla_w \ell(M_k, w_k, \xi_k) \mid \xi_k]\rangle] \\
&= 0.
\end{aligned}
\tag{33}
$$

For the fifth term of (29), we have

$$
\begin{aligned}
&2\langle w_k - w^\star(M_k) - \beta_k \nabla_w f(M_k, w_k), w^\star(M_k) - w^\star(M_{k+1})\rangle \\
&\leq \frac{\lambda \beta_k}{2} \|w_k - w^\star(M_k) - \beta_k \nabla_w f(M_k, w_k)\|^2 + \frac{2}{\lambda \beta_k} \|w^\star(M_k) - w^\star(M_{k+1})\|^2 \\
&\leq \frac{\lambda \beta_k}{2}(1 - \frac{3\lambda \beta_k}{2})\|w_k - w^\star(M_k)\|^2 + \frac{2L^2(L_\Phi^2 + \sigma^2)\alpha_k^2}{\lambda^3 \beta_k} \\
&\leq \frac{\lambda \beta_k}{2}\|w_k - w^\star(M_k)\|^2 + \frac{2L^2(L_\Phi^2 + \sigma^2)\alpha_k^2}{\lambda^3 \beta_k},
\end{aligned}
\tag{34}
$$

where the second inequality follows from (30) and (32).

Collecting the bounds from (30)-(34) and plugging them into (29),

$$
\begin{aligned}
&\mathbb{E}[\|w_{k+1} - w^\star(M_{k+1})\|^2] \\
&\leq (1 - \frac{3\lambda \beta_k}{2})\mathbb{E}[\|w_k - w^\star(M_k)\|^2] + 2\sigma^2 \beta_k^2 + \frac{L^2(L_\Phi^2 + \sigma^2)\alpha_k^2}{\lambda^2} \\
&\quad + \frac{\lambda \beta_k}{2}\mathbb{E}[\|w_k - w^\star(M_k)\|^2] + \frac{2L^2 \alpha_k^2}{\lambda^3 \beta_k}\mathbb{E}[\|\nabla \Phi_{1/\hat{\rho}}(M_k)\|^2] + \frac{2L^2(L_\Phi^2 + \sigma^2)\alpha_k^2}{\lambda^3 \beta_k} \\
&\leq (1 - \lambda \beta_k)\mathbb{E}[\|w_k - w^\star(M_k)\|^2] + \frac{4L^2(L_\Phi^2 + \sigma^2)\alpha_k^2}{\lambda^3 \beta_k} + 2\sigma^2 \beta_k^2,
\end{aligned}
$$

where the second inequality follows from the step size condition $\beta_k \leq \frac{2}{\lambda}$.

$\square$

## C.3. Proof of Proposition B.3

By the update rule of $M_k$ in (2),

$$
\begin{aligned}
&\|M_{k+1} - M^\star\|^2 \\
&= \|\operatorname{proj}_{\mathcal{M}}(M_k - \alpha_k G_M(M_k, w_k, \xi_k)) - M^\star\|^2 \\
&\leq \|M_k - \alpha_k G_M(M_k, w_k, \xi_k) - M^\star\|^2 \\
&= \left\| M_k - M^\star - \alpha_k G_M(M_k, w^\star(M_k), \xi_k) + \alpha_k\Big(G_M(M_k, w^\star(M_k), \xi_k) - G_M(M_k, w_k, \xi_k)\Big) \right\|^2 \\
&= \|M_k - M^\star - \alpha_k G_M(M_k, w^\star(M_k), \xi_k)\|^2 + \alpha_k^2 \|G_M(M_k, w^\star(M_k), \xi_k) - G_M(M_k, w_k, \xi_k)\|^2 \\
&\quad + 2\alpha_k \langle M_k - M^\star - \alpha_k G_M(M_k, w^\star(M_k), \xi_k), G_M(M_k, w^\star(M_k), \xi_k) - G_M(M_k, w_k, \xi_k)\rangle \\
&\leq (1 + \frac{\lambda_\Phi \alpha_k}{4})\|M_k - M^\star - \alpha_k G_M(M_k, w^\star(M_k), \xi_k)\|^2 \\
&\quad + (\alpha_k^2 + \frac{4\alpha_k}{\lambda_\Phi})\|G_M(M_k, w^\star(M_k), \xi_k) - G_M(M_k, w_k, \xi_k)\|^2,
\end{aligned}
\tag{35}
$$

where the first inequality is due to the fact that the projection to a closed convex set is non-expansive.

We bound the first term of (35) in expectation

$$
\begin{aligned}
&\mathbb{E}[\|M_k - M^\star - \alpha_k G_M(M_k, w^\star(M_k), \xi_k)\|^2] \\
&= \mathbb{E}\left[\|M_k - M^\star\|^2 - \alpha_k \langle M_k - M^\star, G_M(M_k, w^\star(M_k), \xi_k)\rangle + \alpha_k^2 \|G_M(M_k, w^\star(M_k), \xi_k)\|^2\right] \\
&= \mathbb{E}[\|M_k - M^\star\|^2] - \alpha_k \mathbb{E}[\langle M_k - M^\star, \mathbb{E}[G_M(M_k, w^\star(M_k), \xi_k) \mid \xi_k]\rangle] \\
&\quad + \alpha_k^2 \mathbb{E}[\|G_M(M_k, w^\star(M_k), \xi_k)\|^2] \\
&= \mathbb{E}[\|M_k - M^\star\|^2] - \alpha_k \mathbb{E}[\langle M_k - M^\star, \nu_k\rangle] + \alpha_k^2 \mathbb{E}[\|\nu_k\|^2] + \alpha_k^2 \mathbb{E}[\|G_M(M_k, w^\star(M_k), \xi_k) - \nu_k\|^2],
\end{aligned}
\tag{36}
$$

where $\nu_k$ is a subgradient of $\Phi$ at $M_k$, i.e. $\nu_k \in \partial\Phi(M_k)$.

The strong convexity of $\Phi$ implies that

$$
\langle M_k - M^\star, \nu_k\rangle \geq \Phi(M_k) - \Phi(M^\star) + \frac{\lambda_\Phi}{2}\|M_k - M^\star\|^2 \geq \frac{\lambda_\Phi}{2}\|M_k - M^\star\|^2.
\tag{37}
$$

Combining (36) and (37),

$$
\begin{aligned}
&\mathbb{E}[\|M_k - M^\star - \alpha_k G_M(M_k, w^\star(M_k), \xi_k)\|^2] \\
&\leq \mathbb{E}[\|M_k - M^\star\|^2] - \frac{\lambda_\Phi \alpha_k}{2}\mathbb{E}[\|M_k - M^\star\|^2] + \alpha_k^2 \mathbb{E}[\|\nu_k\|^2] + \alpha_k^2 \mathbb{E}[\|G_M(M_k, w^\star(M_k), \xi_k) - \nu_k\|^2] \\
&\leq (1 - \frac{\lambda_\Phi \alpha_k}{2})\mathbb{E}[\|M_k - M^\star\|^2] + (L_\Phi^2 + \sigma^2)\alpha_k^2,
\end{aligned}
\tag{38}
$$

where the second inequality is due to Assumption 3.4 and Lemma A.2.

The second term of (35) can be bounded using the Lipschitz continuity of $G_M$ from Assumption 3.5

$$
\|G_M(M_k, w^\star(M_k), \xi_k) - G_M(M_k, w_k, \xi_k)\|^2 \leq L^2 \|w_k - w^\star(M_k)\|^2.
\tag{39}
$$

Combining (35), (38), and (39),

$$
\begin{aligned}
&\mathbb{E}[\|M_{k+1} - M^\star\|^2] \\
&\leq (1 + \frac{\lambda_\Phi \alpha_k}{4})\Big((1 - \frac{\lambda_\Phi \alpha_k}{2})\mathbb{E}[\|M_k - M^\star\|^2] + (L_\Phi^2 + \sigma^2)\alpha_k^2\Big) + (\alpha_k^2 + \frac{4\alpha_k}{\lambda_\Phi})\mathbb{E}[L^2 \|w_k - w^\star(M_k)\|^2] \\
&\leq (1 - \frac{\lambda_\Phi \alpha_k}{4})\mathbb{E}[\|M_k - M^\star\|^2] + \frac{(4 + \lambda_\Phi)L^2 \alpha_k}{\lambda_\Phi}\mathbb{E}[\|w_k - w^\star(M_k)\|^2] + \frac{(4 + \lambda_\Phi)(L_\Phi^2 + \sigma^2)\alpha_k^2}{4},
\end{aligned}
$$

where the second inequality follows from the step size condition $\alpha_k \leq 1$.

$\square$

# D. Proof of Lemmas

## D.1. Proof of Lemma 3.2

For any fixed $D$, we have $\forall t > 0$

$$\Phi(M + tD) - \Phi(M) = f(M + tD, w^\star(M + tD)) - f(M, w^\star(M))$$
$$\leq f(M + tD, w^\star(M)) - f(M, w^\star(M)). \tag{40}$$

Recall the definition of Clarke directional derivative in (14)-(15). Dividing (40) by $t$ and taking the limsup,

$$\Phi^\circ(M; D) \leq f_M^\circ(M, w^\star(M); D). \tag{41}$$

Define the shorthand notation $w_t = w^\star(M + tD)$. By the definition of $w^\star$,

$$\Phi(M + tD) - \Phi(M) = f(M + tD, w_t) - f(M, w^\star(M))$$
$$= \Big(f(M + tD, w_t) - f(M, w_t)\Big) + \Big(f(M, w_t) - f(M, w^\star(M))\Big)$$
$$\geq f(M + tD, w_t) - f(M, w_t).$$

Again, dividing by $t$ and taking the limsup,

$$\Phi^\circ(M; D) \geq \limsup_{t \downarrow 0} \frac{f(M + tD, w_t) - f(M, w_t)}{t}. \tag{42}$$

We now argue that $w_t \to w^\star(M)$. Since $\mathcal{W}$ is compact, $w_t$ has accumulation points. Suppose that along some sequence $t_k \downarrow 0$, we have $w_{t_k} \to \bar{w}$. We need to show $f(M, \bar{w}) = \Phi(M)$. The following relationship follows from the continuity of $f$.

$$f(M, \bar{w}) = \lim_{k \to \infty} f(M, w_{t_k}) = \lim_{k \to \infty} f(M + t_k D, w_{t_k}) = \lim_{k \to \infty} \Phi(M + t_k D) = \Phi(x). \tag{43}$$

The uniqueness of $w^\star(M)$ together with (43) implies that $w_t \to w^\star(M)$. As a result, we have

$$\limsup_{t \downarrow 0} \frac{f(M + tD, w_t) - f(M, w_t)}{t} = f^\circ(M, w^\star(M); D).$$

This equation, along with (41) and (42), leads to

$$\Phi^\circ(M; D) = f_M^\circ(M, w^\star(M); D), \quad \forall D.$$

The claimed result follows in light of (16).

$\square$

## D.2. Proof of Lemma 4.1

We prove by providing an instance of such functions.

Consider the regression problem with a one-dimensional two-layer neural network and a single data point (i.e. $N = m = n = 1$). Let $\mathcal{M} = \mathcal{W} = [0.01, 10]$ and the activation function be the identity mapping. Suppose that the feature and label are both 1. The regression objective is

$$f(M, w) = (Mw - 1)^2 + 0.1M^2,$$

under a small regularization on the first layer's weight. This function is not jointly convex in $(M, w)$. To see it, we can calculate its Hessian

$$\nabla^2 f(M, w) = \begin{bmatrix} 2w^2 + 0.2 & 4Mw - 2 \\ 4Mw - 2 & 2M^2 \end{bmatrix}.$$

For small values of $(M, w)$, this matrix has negative eigenvalues (one can verify at, for example, $M = w = 0.1$ or $M = w = 0.5$). The function $f$ is not convex as its Hessian is not positive semi-definite.

We then compute the reduced single-variable objective with respect to $M$. Since $w^\star(M) = \min(10, 1/M)$, we have

$$\Phi(M) = \left(M \cdot \min(10, 1/M) - 1\right)^2 + 0.1M^2.$$

Since $(M \cdot \min(10, 1/M) - 1)^2$ is convex in $M$ on $\mathcal{M}$ (easily verifiable by visualization), $\Phi(M)$ is 0.05-strongly convex.

$\square$

### D.3. Proof of Lemma 5.1

We verify the assumptions one by one.

**Verification of Assumption 3.1.** The Hessian of $f(M, \cdot)$ is

$$\nabla_w^2 f(M, w) = 2\phi(XM)^\top \phi(XM) + \lambda I.$$

Therefore, we have

$$\nabla_w^2 f(M, w) \succeq 2\lambda I,$$

which means that $f(M, \cdot)$ is $\lambda$-strongly convex.

**Verification of Assumption 3.3.** We know that $w^\star(M)$ satisfies the optimality condition

$$\phi(XM)^\top \left(\phi(XM)w^\star(M) - Y\right) + \lambda w^\star(M) = 0,$$

which yields

$$w^\star(M) = \left(\phi(XM)^\top \phi(XM) + \lambda I\right)^{-1} \phi(XM)^\top Y.$$

Plugging this into $f(M, \omega^\star(M))$ and simplifying the terms, we have the following expression for $\Phi(M)$

$$\Phi(M) = \|Y\|_2^2 - Y^\top \phi(XM)\left(\phi(XM)^\top \phi(XM) + \lambda I\right)^{-1} \phi(XM)^\top Y.$$

Using the Woodbury identity $(I + \frac{1}{\lambda}AA^\top)^{-1} = I - A(A^\top A + \lambda I)^{-1}A^\top$ for any matrix $A$, we obtain the equivalent expression

$$\Phi(M) = Y^\top \left(I + \frac{1}{\lambda}\phi(XM)\phi(XM)^\top\right)^{-1}Y. \tag{44}$$

Drusvyatskiy & Paquette (2019)[Lemma 4.2] establishes that a function $g(x) = h(c(x))$ is weakly convex and Lipschitz if $h$ is convex and $c$ is a smooth map with a Lipschitz Jacobian. In our case, $\left(\phi(XM)^\top \phi(XM) + \lambda I\right)^{-1}$ is always positive definite as $\phi(XM)^\top \phi(XM) + \lambda I$ is positive definite, and the function $A \mapsto Y^\top AY$ is convex and Lipschitz over positive definite matrices. The smoothness of $\phi$ element-wise guarantees the Lipschitz Jacobian of $\left(\phi(XM)^\top \phi(XM) + \lambda I\right)^{-1}$ with respect to $M$. This concludes the proof.

To see a simple example, consider the 1-dimensional case ($N = m = 1$, $M$ and $w$ are scalars) with the feature and label both being 1. Then, $\Phi(M) = \frac{\lambda}{\lambda + \phi(M)^2}$. One can plot this function and easily see that it is not only weakly convex but also smooth.

**Verification of Assumption 3.4.** In Section 5.1 we justify that $G_M$ and $\nabla_w \ell$ are unbiased gradient estimators. The bounded variances follow in a straightforward manner from the boundedness of the feature vectors and the boundedness of $\mathcal{M}, \mathcal{W}$.

**Verification of Assumption 3.5.** Holding $M$ fixed (and bounded due to the boundedness of $\mathcal{M}$), the operator $G_M(M, w, \mathcal{B})$ is quadratic in $w$ and therefore Lipschitz since $\mathcal{W}$ is bounded.

Similarly, $\nabla_w \ell(M, w, \mathcal{B})$ is quadratic (and therefore Lipschitz) in $\phi(X_\mathcal{B} M)$ given that $\phi(X_\mathcal{B} M)$ is bounded (guaranteed by the boundedness of the features and the set $\mathcal{M}$). As the mapping $M \mapsto \phi(X_\mathcal{B} M)$ is also Lipschitz and the composition of Lipschitz mappings is Lipschitz, we can conclude that $\nabla_w \ell(M, w, \mathcal{B})$ is Lipschitz in $M$. The Lipschitz continuity of $\nabla_w \ell(M, w, \mathcal{B})$ with respect to $w$ is simply due to the linearity of $\nabla_w \ell(M, w, \mathcal{B})$ in $w$ and boundedness of $\phi(X_\mathcal{B} M)$.

$\square$

### D.4. Proof of Lemma 5.2

We verify the assumptions one by one.

**Verification of Assumption 3.1.** The logistic loss is convex in its linear argument, and the additional $\ell_2$ regularization term ensures $\lambda$-strong convexity in $w$.

**Verification of Assumption 3.3.** Now that $f$ is differentiable with respect to $M$, recall from Lemma 3.2 that the gradient of $\Phi$ satisfies

$$\nabla \Phi(M) = \nabla_M f(M, w^\star(M)). \tag{45}$$

Due to the Lipschitz and smooth activation function, $f$ is the composition of Lipschitz, smooth operators and is therefore Lipschitz and smooth. Together with (45), this implies that $\nabla \Phi$ is Lipschitz in $M$ as long as $w^\star$ is Lipschitz. The Lipschitz continuity of $w^\star$ is guaranteed under the strong convexity of $f$ with respect to $w$, which we show in Lemma A.1.

Note that smoothness is a stronger condition than weak convexity. Since $f$ is smooth, i.e. has Lipschitz gradients, it is also weakly convex.

**Verification of Assumption 3.4.** As we explain in Section 5.2, the operators $G_M, \nabla_w \ell$ return unbiased stochastic (sub)gradient estimators by construction. Boundedness of $x_i$, together with boundedness of $\mathcal{M}$ and $\mathcal{W}$, implies bounded second moments.

**Verification of Assumption 3.5.** To show $G_M(M, w, \mathcal{B})$ is Lipschitz in $w$, it suffices to show that $s(\phi(XM)w)w^\top$ is Lipschitz. The Lipschitz continuity of $s(\phi(XM)w)w^\top$ follows from the fact that $s$ (the sigmoid function) is Lipschitz and that $\mathcal{W}$ is bounded. A similar argument can be used to show that $G_M(M, w, \mathcal{B})$ is Lipschitz in $M$ and $w$.

$\square$

### D.5. Proof of Lemma A.1

Due to the definition of $w^\star$, we have $\forall w$

$$\langle \nabla_w f(M, w^\star(M)), w - w^\star(M) \rangle \geq 0, \quad \langle \nabla_w f(M', w^\star(M')), w - w^\star(M') \rangle \geq 0,$$

which implies

$$\langle \nabla_w f(M, w^\star(M)), w^\star(M') - w^\star(M) \rangle \geq 0, \quad \langle \nabla_w f(M', w^\star(M')), w^\star(M) - w^\star(M') \rangle \geq 0,$$

Adding the two inequalities yields

$$\langle \nabla_w f(M', w^\star(M')) - \nabla_w f(M, w^\star(M)), w^\star(M') - w^\star(M) \rangle \leq 0.$$

Rearranging the terms, we get

$$\langle \nabla_w f(M, w^\star(M')) - \nabla_w f(M, w^\star(M)), w^\star(M') - w^\star(M) \rangle$$
$$\leq \langle \nabla_w f(M, w^\star(M')) - \nabla_w f(M', w^\star(M')), w^\star(M') - w^\star(M) \rangle. \tag{46}$$

We upper bound the right hand side of (46) under Assumption 3.5

$$\langle \nabla_w f(M, w^\star(M')) - \nabla_w f(M', w^\star(M')), w^\star(M') - w^\star(M) \rangle$$
$$\leq \|\nabla_w f(M, w^\star(M')) - \nabla_w f(M', w^\star(M'))\| \|w^\star(M') - w^\star(M)\|$$
$$\leq \|\mathbb{E}[\nabla_w \ell(M, w^\star(M'), \xi) - \nabla_w \ell(M', w^\star(M'), \xi)]\| \|w^\star(M') - w^\star(M)\|$$
$$\leq \mathbb{E}[\|\nabla_w \ell(M, w^\star(M'), \xi) - \nabla_w \ell(M', w^\star(M'), \xi)\|] \|w^\star(M') - w^\star(M)\|$$
$$\leq L\|M - M'\| \|w^\star(M') - w^\star(M)\|. \tag{47}$$

The $\lambda$-strong convexity of $f(M, \cdot)$ allows us to lower bound the left hand side of (46)

$$\langle \nabla_w f(M, w^\star(M')) - \nabla_w f(M, w^\star(M)), w^\star(M') - w^\star(M) \rangle \geq \lambda \|w^\star(M') - w^\star(M)\|^2. \tag{48}$$

Combining (46)-(48),

$$\lambda \|w^\star(M') - w^\star(M)\|^2 \leq L\|M - M'\| \|w^\star(M') - w^\star(M)\|,$$

which implies

$$\|w^\star(M') - w^\star(M)\| \leq \frac{L}{\lambda} \|M - M'\|. \tag{49}$$

$\square$

### D.6. Proof of Lemma A.2

By Assumption 3.5,

$$|\Phi(M) - \Phi(M')| = |f(M, w^\star(M)) - f(M', w^\star(M'))|$$
$$\leq L\|M - M'\| + L\|w^\star(M) - w^\star(M')\|$$
$$\leq L\|M - M'\| + L \cdot \frac{L}{\lambda} \|M - M'\|$$
$$= \frac{L(\lambda + 1)}{\lambda} \|M - M'\|. \tag{50}$$

The Lipschitz continuity of $\Phi$ implies its Clarke subgradients are all bounded. To see this, note that for any $t > 0$ and direction $D$, we have from (50)

$$\left| \frac{\Phi(M + tD) - \Phi(M)}{t} \right| \leq \frac{L(\lambda + 1)}{\lambda} \|D\|,$$

which, in light of (14), implies the following inequality

$$\Phi^\circ(M; D) \leq \frac{L(\lambda + 1)}{\lambda} \|D\|$$

.

Then, by the definition of $\partial \Phi$ in (16), we have for all $\nu \in \partial \Phi(M)$ such that $\|\nu\| \neq 0$

$$\nu^\top D \leq \Phi^\circ(M; D) \leq \frac{L(\lambda + 1)}{\lambda} \|D\|.$$

Choosing $D = \nu/\|\nu\|$ yields

$$\|\nu\| \leq \frac{L(\lambda + 1)}{\lambda}.$$

$\square$

# E. Gradient-Based Temporal Difference Learning Under Neural Network Function Approximation

We first present the expressions of the gradients the MSPBE and then make the derivation.

$$
\begin{aligned}
\nabla_M f(M, w) = 2\mathbb{E}_\pi\Big[&\gamma\big(\psi_M(s)^\top \mu(M,w)\big)\nabla_M\big(\psi_M(s')^\top w\big) - \big(\psi_M(s)^\top \mu(M,w)\big)\nabla_M\big(\psi_M(s)^\top w\big) \\
&+ \big(r(s,a) + \gamma\psi_M(s')^\top w - \psi_M(s)^\top w - \psi_M(s)^\top \mu(M,w)\big)\nabla_M\big(\psi_M(s)^\top w\big)\Big]
\end{aligned}
$$

$$
\nabla_w f(M,w) = -2\mathbb{E}_\pi\Big[\big(r(s,a) + \gamma\psi_M(s')^\top w - \psi_M(s)^\top w\big)\psi_M(s)\Big] + 2\gamma\mathbb{E}_\pi[\psi_M(s')\psi_M(s)^\top]\mu(M,w),
$$

**Gradient Derivation.** Sutton et al. (2009) shows that the MSPBE objective can be alternatively expressed as

$$
\begin{aligned}
f(M,w) &= \mathbb{E}[\delta_{M,w}(s,a,s')\psi_M(s)]^\top \mathbb{E}[\psi_M(s)\psi_M(s)^\top]^{-1}\mathbb{E}[\delta_{M,w}(s,a,s')\psi_M(s)] \\
&= b(M,w)^\top C(M)^{-1}b(M,w),
\end{aligned}
$$

where we define $\delta_{M,w}(s,a,s') \triangleq r(s,a) + \gamma\psi_M(s')^\top w - \psi_M(s)^\top w$, $b(M,w) \triangleq \mathbb{E}[\delta_{M,w}(s,a,s')\psi_M(s)]$, and $C(M) \triangleq \mathbb{E}[\psi_M(s)\psi_M(s)^\top]$.

With $M$ fixed, the goal of $w$ is to solve the standard TDC problem under (fixed) linear function approximation, where the feature vector for state $s$ is $\psi_M(s)$. This observation allows us to directly invoke the derivation in Sutton et al. (2009)[Section 4] and conclude

$$
\nabla_w f(M,w) = -2\mathbb{E}_\pi\Big[\big(r(s,a) + \gamma\psi_M(s')^\top w - \psi_M(s)^\top w\big)\psi_M(s)\Big] + 2\gamma\mathbb{E}_\pi[\psi_M(s')\psi_M(s)^\top]\mu(M,w).
$$

What remains is to derive the gradient with respect to $M$. Under any variation $dM$, we have

$$
\begin{aligned}
df &= (db)^\top C^{-1}b + b^\top d(C^{-1})b + b^\top C^{-1}db \\
&= 2(db)^\top C^{-1}b - b^\top C^{-1}(dC)C^{-1}b \\
&= 2(db)^\top \mu - \mu^\top(dC)\mu,
\end{aligned}
$$

where the second equation follows from the standard identity $d(C^{-1}) = -C^{-1}(dC)C^{-1}$, and in the third equation we define $\mu(M,w) = C(M)^{-1}b(M,w)$.

By the definition of $b$ and $C$,

$$
db = \mathbb{E}[(d\delta_{M,w})\psi_M + \delta_{M,w}d\psi_M], \quad dC = \mathbb{E}[d\psi_M\psi_M^\top + \psi_M d\psi_M^\top]
$$

Plugging $db, dC$ into $df$,

$$
\begin{aligned}
df &= 2\mathbb{E}[(d\delta_{M,w})\psi_M + \delta_{M,w}d\psi_M]^\top \mu - \mu^\top \mathbb{E}[d\psi_M\psi_M^\top + \psi_M d\psi_M^\top]\mu \\
&= 2\mathbb{E}[(\psi_M^\top \mu)d\delta_{M,w}] + 2\mathbb{E}[(d\psi_M)^\top \mu\delta_{M,w}] - \mathbb{E}[\mu^\top d\psi_M\psi_M^\top \mu] - \mathbb{E}[\mu^\top \psi_M(d\psi_M)^\top \mu] \\
&= 2\mathbb{E}[(\psi_M^\top \mu)d\delta_{M,w}] + 2\mathbb{E}[(d\psi_M)^\top \mu\delta_{M,w}] - 2\mathbb{E}[(d\psi_M)^\top \mu\big(\psi_M^\top \mu\big)] \\
&= 2\mathbb{E}\left[(\psi_M^\top \mu)d\delta_{M,w} + \big(\delta_{M,w} - \psi_M^\top \mu\big)(d\psi_M)^\top \mu\right].
\end{aligned}
$$

Note that $d\delta_{M,w} = \gamma d\psi_M(s')^\top w - d\psi_M(s)^\top w$, and that for a vector $v \in \mathbb{R}^n$ independent of $M$ we have $(d\psi_M)^\top v = \langle\nabla_M(\psi_M^\top v), dM\rangle$. These relationships imply

$$
\begin{aligned}
\nabla_M f(M,w) = 2\mathbb{E}\Big[&\gamma(\psi_M(s)^\top \mu(M,w))\nabla_M(\psi_M(s')^\top w) - (\psi_M(s)^\top \mu(M,w))\nabla_M(\psi_M(s)^\top w) \\
&+ \big(r(s,a) + \gamma\psi_M(s')^\top w - \psi_M(s)^\top w - \psi_M(s)^\top \mu(M,w)\big)\nabla_M(\psi_M(s)^\top w)\Big].
\end{aligned}
$$

**Algorithm.** Given current iterates $M_k, w_k$, we can estimate $\mu(M_k, w_k)$ by maintaining an estimate $\mu_k$ and updating $\mu_k$ in the same loop along with $M_k, w_k$. The formal updates are given in Algorithm 1. Note that (51) is uses standard stochastic to solve for the fixed point equation (13) under $M_k, w_k$.

The algorithm involves three simultaneous updates. We know from the existing literature on multi-sequence stochastic approximation (Shen & Chen, 2022) that we can choose $\zeta_k$ to decay equally fast with respect to $k$ as $\beta_k$ (up to a multiplicative factor difference) and extend our analysis to still guarantee the $\mathcal{O}(k^{-2/3})$ convergence rate.

The requirement on sampling i.i.d. from $d_\pi$ can also be replaced by Markovian sampling according to the state transition. It is a very well-known result in the literature that dependent Markovian samples only incur an additional logarithm factor in the convergence rate under the standard geometric mixing rate assumption (Srikant & Ying, 2019; Chen et al., 2022).

---

**Algorithm 1** Temporal Difference Learning with Gradient Correction (Neural Network Function Approximation)

---

1: **Initialize:** Body-layer parameters $M_0$, final-layer parameters $w_0$, auxiliary variable $\mu_0$
2: **for** iteration $k = 0, 1, 2, \dots$ **do**
3:     Sample $s_k \sim d_\pi, a \sim \pi(\cdot \mid s_k), s'_k \sim \mathcal{P}(\cdot \mid s_k, a_k)$, observe reward $r(s_k, a_k)$
4:     Compute TD error

$$\delta_k = r(s_k, a_k) + \gamma\psi_{M_k}(s'_k)^\top w_k - \psi_{M_k}(s_k)^\top w_k$$

5:     Update body-layer parameters

$$M_{k+1} = M_k - 2\alpha_k\Big(\gamma\psi_{M_k}(s_k)^\top \mu_k \nabla_M(\psi_{M_k}(s'_k)^\top w_k) - \psi_{M_k}(s_k)^\top \mu_k \nabla_M(\psi_{M_k}(s_k)^\top w_k)$$
$$+ (\delta_k - \psi_{M_k}(s_k)^\top \mu_k)\nabla_M(\psi_{M_k}(s_k)^\top w_k)\Big)$$

6:     Update final-layer parameters

$$w_{k+1} = w_k - 2\beta_k\Big(\gamma\psi_{M_k}(s'_k)\psi_{M_k}(s_k)^\top \mu_k - \delta_k\psi_{M_k}(s_k)\Big)$$

7:     Update auxiliary variable

$$\mu_{k+1} = \mu_k - \zeta_k\Big(\psi_{M_k}(s_k)\psi_{M_k}(s_k)^\top \mu_k - \delta_k\psi_{M_k}(s)\Big) \tag{51}$$

8: **end for**

---

**Sketch of Convergence Analysis of TDC Under Neural Network Function Approximation.** We provide a proof sketch of the convergence of TDC in the presence of the auxiliary variable $\mu_k$ under i.i.d. samples. The proof of two-time-scale (and multi-time-scale) algorithms typically proceeds by first establishing the iteration-wise drift of the fast and slow variables individually and then combining their iteration-wise bounds through a proper Lyapunov function. The proofs of Theorems 3.6 and 3.7 follow this approach, and the analysis of TDC proceeds similarly as follows.

Recall the definition of $\mu(M, w)$ in Eq.(13). To simplify notation, we denote $A(M) = \mathbb{E}_\pi[\psi_M(s)\psi_M(s)^\top]$ and $b(M, w) = \mathbb{E}_\pi[(r(s, a) + \gamma\psi_M(s')^\top w - \psi_M(s)^\top w)\psi_M(s)]$. It is obvious that $A(M)$ is positive semi-definite for all $M$. We further assume that it is positive definite, i.e. there exists a constant $\lambda_A$ such that $A(M) \succeq \lambda_A I$ for all $M$. To find $\mu(M, w)$ we need to solve the (linear) equation

$$A(M)\mu(M, w) - b(M, w) = 0.$$

If $\mu_k$ were estimated exactly accurate in every iteration (meaning that $\mu_k = \mu(M_k, w_k)$), the stochastic gradient of $M_k$ given by line 5 of Algorithm 1 would be an unbiased estimate of the true gradient $\nabla_M f(M, w)$, and the iteration-wise

convergence analysis from Proposition B.1 would apply. However, in reality $\mu_k$ is not exactly accurate, and we instead have

$$\mathbb{E}[\Phi_{1/\hat{\rho}}(M_{k+1})] \leq \mathbb{E}[\Phi_{1/\hat{\rho}}(M_k)] - \frac{(4\hat{\rho}-5\rho)\alpha_k}{4\hat{\rho}}\mathbb{E}[\|\nabla\Phi_{1/\hat{\rho}}(M_k)\|^2]$$
$$+ \frac{L^2\hat{\rho}\alpha_k}{\rho}\mathbb{E}[\|w_k - w^\star(M_k)\|^2] + \frac{(L_\Phi^2+\sigma^2)\hat{\rho}\alpha_k^2}{2} + C_1\alpha_k\mathbb{E}[\|\mu_k - \mu(M_k, w_k)\|^2],$$

where the last term is the only difference from the bound in Proposition B.1 capturing the optimality gap of $\mu_k$. Here $C_1$ is a constant depending on certain Lipschitz and boundedness parameters.

Similarly, the iteration-wise convergence of $w_k$ can be established extending the result of Proposition B.2.

$$\mathbb{E}[\|w_{k+1} - w^\star(M_{k+1})\|^2]$$
$$\leq (1 - \lambda\beta_k)\mathbb{E}[\|w_k - w^\star(M_k)\|^2] + \frac{4L^2(L_\Phi^2+\sigma^2)\alpha_k^2}{\lambda^3\beta_k} + 2\sigma^2\beta_k^2 + C_2\beta_k\mathbb{E}[\|\mu_k - \mu(M_k, w_k)\|^2],$$

where $C_2$ is again a constant.

It can be shown that the iteration-wise convergence of $\mu_k$ admits the following bound. The argument used to establish the bound is standard and resembles that in Zeng et al. (2025)[Section C.4].

$$\mathbb{E}[\|\mu_{k+1} - \mu(M_{k+1}, w_{k+1})\|^2] \leq (1 - C_3\lambda_A\zeta_k)\mathbb{E}[\|\mu_k - \mu(M_k, w_k)\|^2] + \frac{C_4\beta_k^2}{\zeta_k}\mathbb{E}[\|w_k - w^\star(M_k)\|^2]$$
$$+ \frac{C_4\alpha_k^2}{\zeta_k} + C_6\zeta_k^2.$$

Combining the three inequalities above (with the bound on $\|w_{k+1} - w^\star(M_{k+1})\|^2$ scaled by $\frac{8L^2\hat{\rho}^2\alpha_k}{(4\hat{\rho}-5\rho)\rho\lambda\beta_k}$ and the bound on $\|\mu_{k+1} - \mu(M_{k+1}, w_{k+1})\|^2$ scaled by $\frac{C_7\alpha_k}{\zeta_k}$ where $C_7 = \max\{\frac{2L^2\hat{\rho}}{\rho C_1 C_3\lambda_A}, \frac{8\hat{\rho}C_1}{C_3(4\hat{\rho}-5\rho)\lambda_A} + \frac{16L^2\hat{\rho}^2 C_2}{C_3(4\hat{\rho}-5\rho)\rho\lambda\lambda_A}\}$), we have

$$\alpha_k\mathbb{E}[\|\nabla\Phi_{1/\hat{\rho}}(M_k)\|^2]$$
$$\leq \frac{4\hat{\rho}}{(4\hat{\rho}-5\rho)}\mathbb{E}[\Phi_{1/\hat{\rho}}(M_k) - \Phi_{1/\hat{\rho}}(M_{k+1})] + \frac{4\hat{\rho}C_1\alpha_k}{(4\hat{\rho}-5\rho)}\mathbb{E}[\|\mu_k - \mu(M_k, w_k)\|^2]$$
$$+ \frac{8L^2\hat{\rho}^2\alpha_k}{(4\hat{\rho}-5\rho)\rho\lambda\beta_k}\mathbb{E}[\|w_k - w^\star(M_k)\|^2 - \|w_{k+1} - w^\star(M_{k+1})\|^2] - \frac{4L^2\hat{\rho}^2\alpha_k}{(4\hat{\rho}-5\rho)\rho}\mathbb{E}[\|w_k - w^\star(M_k)\|^2]$$
$$+ \frac{8L^2\hat{\rho}^2 C_2\alpha_k}{(4\hat{\rho}-5\rho)\rho\lambda}\mathbb{E}[\|\mu_k - \mu(M_k, w_k)\|^2]$$
$$+ \frac{C_7\alpha_k}{\zeta_k}\mathbb{E}[\|\mu_k - \mu(M_k, w_k)\|^2 - \|\mu_{k+1} - \mu(M_{k+1}, w_{k+1})\|^2] - \frac{C_3 C_7\lambda_A\alpha_k}{2}\mathbb{E}[\|\mu_k - \mu(M_k, w_k)\|^2]$$
$$+ \frac{2L^2\hat{\rho}C_4\alpha_k\beta_k^2}{\rho C_1 C_3\lambda_A\zeta_k^2}\mathbb{E}[\|w_k - w^\star(M_k)\|^2]$$
$$+ \mathcal{O}\left(\alpha_k^2 + \alpha_k\beta_k + \alpha_k\zeta_k + \frac{\alpha_k^3}{\beta_k^2} + \frac{\alpha_k^3}{\zeta_k^2}\right).$$

Note that the sum of the blue highlighted terms are non-positive by the definition of $C_7$. Choose the step sizes as

$$\alpha_k = \frac{\alpha_0}{(k+1)^{3/5}}, \quad \beta_k = \frac{\beta_0}{(k+1)^{2/5}}, \quad \zeta_k = \frac{\zeta_0}{(k+1)^{2/5}}$$

with $\frac{\beta_0}{\zeta_0} \leq \sqrt{\frac{2C_1 C_3\lambda_A\hat{\rho}}{(4\hat{\rho}-5\rho)C_4}}$. Then, the sum of the red highlighted terms are also non-positive, and the inequality above simplifies to

$$\alpha_k\mathbb{E}[\|\nabla\Phi_{1/\hat{\rho}}(M_k)\|^2]$$
$$\leq \frac{4\hat{\rho}}{(4\hat{\rho}-5\rho)}\mathbb{E}[\Phi_{1/\hat{\rho}}(M_k) - \Phi_{1/\hat{\rho}}(M_{k+1})] + \frac{8L^2\hat{\rho}^2\alpha_k}{(4\hat{\rho}-5\rho)\rho\lambda\beta_k}\mathbb{E}[\|w_k - w^\star(M_k)\|^2 - \|w_{k+1} - w^\star(M_{k+1})\|^2]$$
$$+ \frac{C_7\alpha_k}{\zeta_k}\mathbb{E}[\|\mu_k - \mu(M_k, w_k)\|^2 - \|\mu_{k+1} - \mu(M_{k+1}, w_{k+1})\|^2] + \mathcal{O}\left(\alpha_k^2 + \alpha_k\beta_k + \alpha_k\zeta_k + \frac{\alpha_k^3}{\beta_k^2} + \frac{\alpha_k^3}{\zeta_k^2}\right).$$

The rest of the analysis proceeds in a way identical to the proof of Theorem 3.6 after the step size specification.

It should not be considered a surprising result that the convergence rate of Algorithm 1 is the same as that of (2) in the presence of an additional variable $\mu_k$ (updated on the fastest time scale with step size $\zeta_k$). It is known in the literature of multi-time-scale stochastic approximation that additional fast-time-scale variable(s) can be solved "for free" (i.e. without slowing down the overall convergence rate) if the operator associated with the variable is linear or strongly monotone (Shen & Chen, 2022; Zeng & Doan, 2024; Huang et al., 2025).

