# OpenReview forum: "Rethinking Neural Network Learning Rates: A Stackelberg Perspective"
_ICML.cc/2026/Conference — ICML 2026 regular_

### Official Review · Reviewer_j8i5 · 2026-02-15

**Soundness:** 3
**Presentation:** 4
**Significance:** 3
**Originality:** 3
**Overall Recommendation:** 4
**Confidence:** 4

**Summary:**

The paper formalizes the empirical observation that non-uniform learning rates (slow body, fast head) can accelerate neural network training. They show this is equivalent to two-timescale alternating SGD on a Stackelberg reformulation of the training objective, where the head rapidly best-responds to slowly-evolving body features. They prove finite-time convergence $O(k^{-2/5})$ generally, $O(k^{-2/3})$ under strong convexity of the Stackelberg objective) under broad conditions including non-smooth activations and constraint sets. Two acceleration mechanisms are identified: (i) the Stackelberg reformulation can be strongly convex when the joint objective is not, and (ii) the Stackelberg objective exhibits sharper curvature, providing stronger gradient signals in early training. Experiments section include regression, classification, and RL.

**Compliance With Llm Reviewing Policy:**

Affirmed.

**Key Questions For Authors:**

Here are a few points for the authors to consider.

1. The experimental evaluation, while broad, is somewhat shallow. The regression benchmarks (Friedman, Boston Housing) are toy-scale. The classification experiments (MNIST, Fashion-MNIST, CIFAR-10) are standard but not large-scale. The absence of any ImageNet-scale or language modeling experiment is a significant gap for a paper making claims about “modern neural networks.” The Atari experiments are interesting but, as the authors acknowledge, fall outside the scope of their theory.

2. The paper uses RMSProp optimizer with constant step sizes in experiments while the theory requires SGD with diminishing step sizes. This gap, although not a significant point, is worth addressing. It would be reassuring to see at least one experiment that actually uses the algorithm analyzed in the theory (SGD with diminishing rates and the prescribed $\alpha/\beta$ ratio).

3. The “sharper curvature” mechanism is demonstrated through a 3D landscape visualization of a synthetic regression objective, which is illustrative but not convincing for deep networks. Does this curvature enhancement persist in high-dimensional, non-convex settings? Some empirical measurement of the Stackelberg objective’s condition number or Hessian spectrum during actual deep network training would strengthen this claim greatly.

4. The comparison framework between $(\alpha, \beta)$ non-uniform and uniform slow and fast is sound, but it doesn’t compare against other principled approaches like LARS, LAMB, or the closed-form last-layer methods referred to in the related works section. It would be great to see if additional comparisons can be added.

**Limitations:**

Yes.

**Strengths And Weaknesses:**

This is a clean, well-structured theory paper with a satisfying central insight (Stackelberg reformulation + emergent strong convexity). The theoretical contributions are the main strength. The experiments support the theory and span a great breadth of areas.

---

> ### Author Rebuttal · Authors · 2026-03-30
>
> We thank the reviewer for the time, effort, and valuable feedback. We also thank the reviewer for recognizing our technical contributions. We provide comment-by-comment responses below, and have made the corresponding revisions in the updated paper. Due to the space limit, we cannot respond to question 4. We would greatly appreciate it if the reviewer could kindly refer to our additional response, which we will post once the author–reviewer discussion period begins.
>
> 1. **Reviewer's comment:**
>
> >The experimental evaluation, while broad, is somewhat shallow...
>
> **Response:**
> We agree with the reviewer that the experimental results presented in our work alone does not fully support the claim that non-uniform learning rates leads to improved performance. However, note that empirical evidence for the effectiveness of non-uniform learning rates on large-scale modern neural networks (some of which we cited in the introduction) already exists in the literature and is exactly what motivated our work. Our contribution is complementary to these past empirical studies: rather than presenting a new large-scale empirical finding, we provide a theoretical framework that helps explain why and when non-uniform learning rates can be beneficial. The purpose of our experiments is to broadly validate the effect of non-uniform learning rates, covering a range of problem classes from supervised learning to RL.
>
> Regarding the Atari experiments, our goal is to evaluate non-uniform learning rates in regimes that we cannot rigorously analyze so far. While deep Q learning falls outside the scope of our current analysis, the experiments test whether the observed benefits persist more broadly. Encouragingly, the empirical results indicate that non-uniform learning rates can provide consistent benefits even in these more complex settings, an observation that may help motivate and guide the development of more general theoretical frameworks in the future.
>
> 2. **Reviewer's comment:**
>
> >The paper uses RMSProp optimizer with constant step sizes in experiments while the theory requires SGD with diminishing step sizes...
>
> **Response:**
> The reviewer is correct that the optimizer used in the experiments in Section 5 is RMSProp rather than SGD. This is a purposeful choice, as we would like to test whether the advantage of non-uniform learning rates is present under optimizers commonly used in practice, beyond the specific SGD optimizer analyzed in theory. However, we note that standard SGD, rather than RMSProp, is used to produce Figure 2, which demonstrates the advantage of non-uniform learning rates in a setting that is directly aligned with our theoretical analysis. We have updated the description of experimental setup in Section 5 and the caption of Figure 2 to explicitly clarify this distinction and avoid potential ambiguity.
>
> 3. **Reviewer's comment:**
>
> >The “sharper curvature” mechanism is demonstrated through a 3D landscape visualization of a synthetic regression objective, which is illustrative but not convincing for deep networks. Does this curvature enhancement persist in high-dimensional, non-convex settings? Some empirical measurement...
>
> **Response:**
> We thank the reviewer for the thoughtful suggestion and agree that additional empirical evidence in high-dimensional, more complex problems would further strengthen the paper. However, a few computational challenges are present.
>
> First, to plot each column of Figure 1, we need to evaluate $F(M_k+\eta_1 d_1+\eta_2 d_2,w_k)$ and $\Phi(M_k+\eta_1 d_1+\eta_2 d_2)$ in a fine grid. The exact evaluation of $F$ and $\Phi$ requires iterating through all samples in the dataset. This quickly becomes computationally expensive as the dataset size and/or network size goes up.
>
> Second and more critically, to evaluate $\Phi(M)$ for a given $M$, we need to know $w^\star(M)$, the final-layer best response. In regression problems with moderate dataset size, $w^\star(M)$ admits a closed-form expression as the solution to a least squares problem, which we can solve using reasonable amount of computation. However, for general learning objectives, such a closed form is unavailable. Approximating $w^\star(M)$ would require running an inner optimization loop over $w$ for each grid point, which becomes computationally infeasible when combined with the already expensive landscape evaluation procedure described above.
>
> That said, we agree that our paper will benefit from visualizations in higher dimension. While extending these results to classification and RL problems remains computationally prohibitive for the reasons discussed above, we are currently working on producing visualizations under the real regression datasets considered in Section 5, using deeper neural networks. Although these datasets are still modest in size, they are substantially larger and more realistic than the synthetic example used in Figure 1, and therefore help bridge the gap towards practical settings.

---

> > ### Author Rebuttal · Reviewer_j8i5 · 2026-04-01
> >
> > Thanks to the authors for addressing my comments, look forward to seeing the subsequent plots on high dimensional neural network optimization.

---

> > > ### Author Response · Authors · 2026-04-02
> > >
> > > We thank the reviewer for acknowledging our response. We only realized after the discussion period began that the system does not allow us to post an additional response until the reviewers respond. Please find below our response to Q4.
> > >
> > > 4. **Reviewer's comment:**
> > >
> > > > The comparison framework between $(\alpha,\beta)$ non-uniform and uniform slow and fast is sound, but it doesn’t compare against other principled approaches like LARS, LAMB, or the closed-form last-layer methods referred to in the related works section. It would be great to see if additional comparisons can be added.
> > >
> > > **Response:**
> > > We thank the reviewer for another thoughtful comment. We believe LARS and LAMB-type methods from You et al. (2017) and You et al. (2020) are technically different from the closed-form last-layer method from Galashov et al. (2025), and we discuss these approaches separately. We have added a discussion of these relevant works in Section 1.1 and the second paragraph of Section 6 in the updated paper.
> > >
> > > LARS and LAMB in spirit are innovative optimizers. Rather than updating the network parameter in the standard gradient direction, these methods rescale the gradient in a layer-wise manner based on the gradient and parameter norms. Therefore, LARS and LAMB are more comparable to other adaptive optimizers like Adam and RMSProp. LARS, LAMB, Adam, and RMSProp still rely on the choice of a base learning rate, which is typically set equal across layers. Our work can be complementary to such studies in the sense that it raises the question of whether the further optimization efficiency may arise from setting non-uniform base learning rates across layers in combination with gradient normalization. This partly motivated us to use the RMSProp optimizer in the experiments in Section 5.
> > >
> > > Regarding closed-form last-layer optimization, we note that such closed-form expressions only exist in the regression setting, and even then require the dataset size to be sufficiently small to make the matrix inversion computationally feasible. While Galashov et al. (2025) discusses a possible way to extend their method to the classification setting, the extension is largely heuristic and requires certain approximation. Moreover, no extension exists for the RL setting considered in our experiments. In regression problems where both closed-form last-layer optimization and SGD with non-uniform learning rates are applicable, we expect the former to have better iteration complexity, since it computes $w^\star(M_k)$ exactly in a single step, whereas our approach approximates $w^\star(M_k)$ via gradient updates on a faster time scale. However, the per-iteration computational cost of closed-form methods is significantly higher due to the need for matrix inversion.
> > >
> > > **References**
> > >
> > > Galashov, A., Da Costa, N., Xu, L., Hennig, P. and Gretton, A., 2025. Closed-Form Last Layer Optimization. arXiv preprint arXiv:2510.04606.
> > >
> > > You, Y., Gitman, I. and Ginsburg, B., 2017. Large batch training of convolutional networks. arXiv preprint arXiv:1708.03888.
> > >
> > > You, Y., Li, J., Reddi, S., Hseu, J., Kumar, S., Bhojanapalli, S., Song, X., Demmel, J., Keutzer, K. and Hsieh, C.J., 2020. Large Batch Optimization for Deep Learning: Training BERT in 76 minutes. In International Conference on Learning Representations.

---

### Official Review · Reviewer_rVbM · 2026-03-06

**Soundness:** 2
**Presentation:** 3
**Significance:** 3
**Originality:** 3
**Overall Recommendation:** 4
**Confidence:** 4

**Summary:**

The paper investigates the training dynamics of neural networks with layer-specific non-uniform learning rates, specifically focusing on the common architectural partition of a feature-extracting "body" and a linear "head". The authors elegantly formulate this as a two-time-scale alternating stochastic gradient descent algorithm solving a Stackelberg optimization problem. By leveraging two-timescale stochastic approximation, they provide finite-time convergence guarantees to first-order stationary points ($\tilde{\mathcal{O}}(k^{-2/5})$ generally, and $\mathcal{O}(k^{-2/3})$ under strong convexity). The theoretical framework also accommodates non-smooth activation functions (e.g., ReLU) using Clarke subdifferentials.

**Compliance With Llm Reviewing Policy:**

Affirmed.

**Final Justification:**

Overall, the rebuttal is helpful and addresses part of my questions, especially by clarifying the scope of the theory. Though some concerns remain insufficiently resolved, particularly those regarding the TDC analysis and its empirical support, I am willing to raise my score to 4.

**Key Questions For Authors:**

Just as a supplement to the weaknesses part: Could the authors clarify the practical impact of the projection operators ($proj_{\mathcal{M}}$ and $proj_{\mathcal{W}}$) used in Equation (2)? Specifically, does the theoretical convergence strictly break down if this bounded constraint is removed, as is standard practice in unconstrained FO-SGD? Alternatively, do the authors argue that this bounded assumption is unlikely to be violated in practice? For instance, under the Neural Tangent Kernel (NTK) / lazy training regime [1] (though infinite width assumption is needed), parameters only vary within a highly localized neighborhood around their initialization. I am not sure whether this NTK insight can be a underlying justification for the bounded parameter space assumption.

Generally speaking, I really enjoyed reading this paper and found the theoretical perspective it offers to be quite insightful. However, the weaknesses outlined above are obvious and unignorable. I am entirely open to raising my score if the authors can adequately address at least some of these core concerns during the rebuttal period.


**Below question is purely for discussion and will not impact the evaluation or score of your submission** (Feel free to skip this entirely if you are constrained by the rebuttal word limit.)

Recently, Google introduced the concept of Nested Learning [2], which has sparked interesting discussions in the community regarding layer-wise update strategies. The core intuition shares similarities with your framework: it posits that different layers in an LLM are responsible for retaining knowledge at different temporal scales. Consequently, some layers require frequent updates (or larger learning rates) to quickly assimilate short-term, transient knowledge, while others require "lazy" updates (or much smaller learning rates) to preserve long-term memory and prevent catastrophic forgetting.

Your work arrives at a similar architectural asymmetry—justifying a larger learning rate for the head and a smaller one for the body—but does so strictly through the rigorous optimization lens of a Stackelberg game rather than a memory-retention perspective. I am curious about your thoughts on this intersection. Do you view the sharper transient curvature and Stackelberg dynamics you identified as the underlying mathematical mechanism that enables the memory compartmentalization seen in Nested Learning? I would love to hear your casual thoughts on how these two paradigms might conceptually align.

[1] Jacot, Arthur, Franck Gabriel, and Clément Hongler. "Neural tangent kernel: Convergence and generalization in neural networks." Advances in neural information processing systems 31 (2018).

[2] Behrouz, Ali, et al. "Nested learning: The illusion of deep learning architectures." arXiv preprint arXiv:2512.24695 (2025).

**Limitations:**

See weaknesses.

**Strengths And Weaknesses:**

## Strengths

1. Framing the layer-wise learning rate disparity as a Stackelberg formulation is a fresh and mathematically rigorous perspective. It provides a formal justification for a practice that is often treated as a mere empirical heuristic.

2. The extension of the analysis to non-smooth activation functions via the Clarke subdifferential and Moreau envelopes is technically sound and aligns well with modern network architectures using ReLUs.

3. The numerical analysis and 3D visualization illustrating how the Stackelberg reduction induces sharper curvature in the transient phase (Figure 1) provide strong intuitive support for the acceleration mechanism.

## Weaknesses
1. To satisfy the Lipschitz continuity of the loss function, the theoretical framework heavily relies on the assumption that the parameter spaces $\mathcal{M}$ and $\mathcal{W}$ are compact, convex sets , and requires a projection step in the update rules (Equation 2). Additionally, the lemmas for specific applications assume bounded input features. In standard, unconstrained FO-SGD training of modern architectures, weights are not typically projected onto bounded domains. This strict bounded constraint creates a gap between the theoretical guarantees and practical unconstrained training.

2. The paper's core premise is the benefit of non-uniform learning rates. However, modern FO-SGD heavily relies on adaptive optimizers (e.g., Adam, AdamW) that naturally implement coordinate-wise non-uniform learning rates via diagonal preconditioners. The experimental section only compares the proposed layer-wise non-uniform RMSProp against baselines using uniform RMSProp. The lack of a baseline using a standard adaptive optimizer (like Adam with a default uniform base learning rate) makes it difficult to assess whether this explicit time-scale separation offers any tangible acceleration or performance benefit over what modern optimizers already achieve implicitly.

3. While the introduction motivates the problem in the context of "modern neural networks" and "deep non-linear feature extractors," the theoretical analysis and the primary supervised learning experiments are heavily restricted to toy, two-layer neural networks and simple datasets.

4. The paper dedicates Section 5.3 and Appendix E to deriving a closed-form, gradient-based TDC algorithm for policy evaluation under neural network function approximation. The authors claim this as a significant theoretical contribution yielding the "first provably convergent variant". However, the experimental section completely completely abandons TDC. Instead, it evaluates DQN on Atari games , an off-policy control setting that the authors explicitly admit "fall outside the scope of our analysis". Consequently, the TDC derivation lacks any empirical validation, and the RL experiments lack theoretical grounding.

5. In the TDC formulation, estimating the exact gradients requires introducing an auxiliary variable $\mu$, which is updated alongside $M$ and $w$. This fundamentally transforms the problem from a two-timescale stochastic approximation (which Theorems 3.6 and 3.7 cover) into a much more complex three-variable coupled system. The authors dismiss this significant challenge by simply stating that "we can choose $\zeta_k$ to decay equally fast... as $\beta_k$ ... and extend our analysis" without providing the actual proof. The interaction and tracking errors between two fast variables ($w$ and $\mu$) and a slow variable ($M$) are non-trivial and cannot be waved away. Furthermore, the paper assumes i.i.d. sampling (Assumption 3.4) , but loosely claims in Appendix E that Markovian sampling "only incur an additional logarithm factor" without proving how the mixing time interacts with their specific non-linear, multi-timescale Stackelberg bounds. For a theoretically focused paper, these gaps are unacceptable.

---

> ### Author Rebuttal · Authors · 2026-03-30
>
> We deeply appreciate the reviewer’s thoughtful and insightful questions and comments. We provide comment-by-comment responses below, and have made the corresponding revisions in the paper. Due to the space limit, we cannot respond to all questions. We would greatly appreciate it if the reviewer could kindly refer to our additional responses, which we will post once the author–reviewer discussion period begins.
>
> 1. **Reviewer's comment:**
>
> >To satisfy the Lipschitz continuity of the loss function, the theoretical framework heavily relies on the assumption...
>
> >Could the authors clarify the practical impact of the projection operators used in Equation (2)...
>
> **Response:**
> We thank the reviewer for this important question. We assume that $\mathcal{M},\mathcal{W}$ are compact (i.e. bounded and closed), convex sets for two main reasons.
>
> First, we formulate the problem in a constrained setting for greater generality. The compactness condition guarantees that the minimizers of $f(M,\cdot)$ and $\Phi$ exist, whereas convexity guarantees that we can reliably find the minimizers. These are minimal conditions commonly imposed in the literature of constrained optimization. The algorithm considered in Eq.(2) is a variant of projected gradient descent. The projection step is important and cannot be removed for constraint satisfaction.
>
> If we consider unconstrained problems and assume that at least a minimizer exists, the assumption that $\mathcal{M},\mathcal{W}$ are convex and closed can be removed. However, as the reviewer correctly pointed out, the Lipschitz continuity of the loss function only holds when the norms of $M$ and $w$ are not arbitrarily large. Therefore, for the purpose of theoretical analysis, we need some form of boundedness, either by assuming that $\|M_k\|,\|w_k\|$ remain bounded for all $k$, or by explicitly enforcing it via projection (equivalently, norm clipping onto an $\ell_2$ ball). We believe that such norm clipping is not restrictive and sometimes carried out in practical neural network training.
>
> We also note that the boundedness assumption (or projection to bounded set) is mainly for theoretical tractability. In practical neural network training, the weights typically do not diverge under reasonable learning rates. As a result, explicit norm clipping or projection is often unnecessary. In fact, we do not perform any such projection in our experiments.
>
> 2. **Reviewer's comment:**
>
> >The paper's core premise is the benefit of non-uniform learning rates. However, modern FO-SGD heavily relies on adaptive optimizers...
>
> **Response:**
> The reviewer makes an insightful comment. We fully agree that modern optimizers usually have learning rate adaptation schemes built-in and that understanding the interplay between non-uniform base learning rates and such adaptation schemes is important. This is exactly the future direction we outlined in the second paragraph of the conclusion section, where the referenced recent paper by Hao et al. presents another (more theoretically grounded) coordinate-wise learning rate adaptation method.
>
> While our theoretical understanding is currently limited in this aspect, our goal in using the RMSProp optimizer in the experiments is to understand the interplay from an experimental perspective. We respectfully correct the reviewer that, like Adam, RMSProp also makes coordinate-wise learning rate adjustment based on a running average of past gradient norms. The difference between Adam and RMSProp lies in the additional use of momentum in Adam. The reason that we chose RMSProp over Adam in the experiments is somewhat casual in nature: RMSProp is arguably easier to understand and justify theoretically, and it is also widely used in practice, only second to Adam and AdamW. In the rebuttal stage, we have replicated the regression and classification experiments with Adam and observed results similar to those under RMSProp. We plan to include the plots across regression, classification, and RL problems under Adam in the appendix in the camera-ready version.
>
> 3. **Reviewer's comment:**
>
> >While the introduction motivates the problem in the context of "modern neural networks" and "deep non-linear feature extractors"...
>
> **Response:**
> We clarify that our theoretical results apply to neural networks of any depth. When there are more than two layers, $M$ includes the parameters of all body layers and $w$ includes the parameters of the last layer. We occasionally focus on two-layer neural networks in the paper to simplify the presentation. Upon revisiting the paper, we realize that certain texts may have given the impression that our results are restricted to the two-layer setting. We have revised such statements and would like to thank the reviewer for helping us improve the clarity. Experimentally, we use small neural networks as the problems are relatively small in nature and can be solved to satisfactory accuracy without resorting to a large architecture.

---

> > ### Author Rebuttal · Reviewer_rVbM · 2026-04-01
> >
> > As the authors’ rebuttal addresses only part of my concerns (W1-3), I would appreciate it if they could respond to the remaining points (W4-5) during the author–reviewer discussion phase.

---

> > > ### Author Response · Authors · 2026-04-01
> > >
> > > **To Q4**
> > >
> > > One practical challenge is that there are no standard benchmarks for policy evaluation. Policy evaluation and policy optimization are closely related, and policy optimization benefits from well-established benchmarks, which provide a more widely accepted testbed for empirical validation.
> > >
> > > In addition, we would like to evaluate non-uniform learning rates in more complex problems than what we can rigorously analyze so far. Encouragingly, experiments suggest consistent benefits in deep Q-learning, which may help motivate more general theoretical frameworks in the future.
> > >
> > > **To Q5**
> > >
> > > The two additional challenges for the analysis of TDC are 1) the auxiliary variable $μ_k$ updated with step size $ζ_k$, and 2) Markovian samples.
> > >
> > > 1) We first handle $μ_k$ under i.i.d. samples.
> > >
> > > Multi-time-scale analysis typically establishes iteration-wise drift for each variable and combines them via a Lyapunov function; we follow this approach.
> > >
> > > If $\mu_k$ were estimated exactly accurate in every iteration (i.e. $μ_k=μ(M_k,w_k)$), Proposition B.1 would apply to $M_k$. However, in reality $μ_k$ is not exactly accurate, and we instead have
> > > \begin{align*}
> > > E[\Phi_{1/\hat{\rho}}(M_{k+1})] &\leq E[\Phi_{1/\hat{\rho}}(M_k)]-\frac{(4\hat{\rho}-5\rho)α_k}{4\hat{\rho}}E[||\nabla\Phi_{1/\hat{\rho}}(M_k)||^2]+\frac{L^2\hat{\rho}α_k}{\rho}E[|| w_k- w^\star(M_k)||^2] + \frac{(L_\Phi^2+\sigma^2)\hat{\rho}α_k^2}{2}+ C_1α_kE[||μ_k-μ(M_k,w_k)||^2],
> > > \end{align*}
> > > where $C_1$ is a constant.
> > >
> > > Similarly, for $w_k$,
> > > \begin{align*}
> > > E[||w_{k+1}-w^\star(M_{k+1})||^2]&\leq (1-λβ_k)E[||w_k- w^\star(M_k)||^2]+\frac{4L^2 (L_\Phi^2+\sigma^2) α_k^2}{λ^3β_k}+2\sigma^2β_k^2+C_2β_kE[||μ_k-μ(M_k,w_k)||^2],
> > > \end{align*}
> > > where $C_2$ is another constant.
> > >
> > > For $μ_k$,
> > > \begin{align*}
> > > E[||μ_{k+1}- μ(M_{k+1},w_{k+1})||^2]&\leq (1-C_3λ_Aζ_k)E[||μ_k-μ(M_k,w_k)||^2]+\frac{C_4β_k^2}{ζ_k}E[||w_k-w^\star(M_k)||^2] +\frac{C_5α_k^2}{ζ_k} +  C_6ζ_k^2,
> > > \end{align*}
> > > where $λ_A>0$ is a lower bound on the eigenvalue of some feature covariance matrix.
> > >
> > > Combining and simplifying yields
> > > $$αE[||\nabla\Phi_{1/\hat{\rho}}(M_k)||^2]\leq\frac{4\hat{\rho}}{(4\hat{\rho}-5\rho)}E[\Phi_{1/\hat{\rho}}(M_k)-\Phi_{1/\hat{\rho}}(M_{k+1})]+ \frac{C_7α_k}{β_k}E[|| w_k- w^\star(M_k)||^2-|| w_{k+1}- w^\star(M_{k+1})||^2]+\frac{C_8α_k}{ζ_k}E[||μ_k-μ(M_k,w_k)||^2-||μ_{k+1}-μ(M_{k+1},w_{k+1})||^2]+ O(α_k^2+α_kβ_k+α_kζ_k+\frac{α_k^3}{β_k^2}+\frac{α_k^3}{ζ_k^2}).$$
> > >
> > > We can choose $α_k=\frac{α_0}{(k+1)^{3/5}},β_k=\frac{β_0}{(k+1)^{2/5}},ζ_k=\frac{ζ_0}{(k+1)^{2/5}}$. The rest of the analysis proceeds the same as the proof of Theorem 3.6 after line 700.
> > >
> > > We note that the argument above for handling $\mu_k$ is not novel. It is known in the literature of multi-time-scale stochastic approximation that additional fast-time-scale variable(s) can be solved "for free" (i.e. without slowing down the overall convergence rate) if the operator associated with the variable is a linear operator or is strongly monotone and satisfies certain smoothness condition (Shen and Chen, 2022; Huang et al., 2025).
> > >
> > > 2) The analysis of stochastic approximation and SGD under Markovian noise is more standard. For brevity, we skip presenting the proof sketch and discuss the known results in the literature.
> > >
> > > Suppose that we analyze the convergence of the standard SGD algorithm
> > > $$x_{k+1}=x_k-α_k \nabla f(x_k,\xi_k),$$
> > > where $\xi_k\in\Xi$ represents the randomness (or sample) and $\nabla f(x,\xi)$ represents a stochastic estimate of the true gradient $\nabla f (x)$ under sample $\xi$. Suppose that the stochastic gradient is unbiased when $\xi$ follows a distribution $D$. Then, if $f$ is strongly convex and the algorithm uses i.i.d. samples $\xi_k\sim D$, it is well known that $x_k$ converges to the optimizer $x^\star$ at rate $O(1/k)$.
> > >
> > > Now, suppose that we do not have access to i.i.d. samples from $D$, but a sampling operator $P$ for which $D$ is the stationary distribution. In other words, if we keep sampling $\xi_{k+1}\sim P(\cdot\mid\xi_k)$, the distribution of $x_k$ will approach $D$ as $k$ increases. Note that this is exactly the sample generation setting of TDC. Under a standard ergodicity assumption, we can show that the SGD algorithm under Markovian samples $\xi_{k+1}\sim P(\cdot\mid\xi_k)$ converges with rate $O(\log k/k)$, different from the rate under i.i.d. samples by a logarithm factor.
> > >
> > > This result is first shown by Srikant and Ying (2019) and later extended to more general function classes and two- and multi-time-scale stochastic approximation. Some notable works include Xu et al. (2019), Wu et al. (2020), Chen et al. (2022).
> > >
> > > **Additional References**
> > >
> > > Chen et al. Finite-sample analysis of nonlinear stochastic approximation with applications in reinforcement learning.
> > >
> > > Huang et al. Single-timescale multi-sequence stochastic approximation without fixed point smoothness.
> > >
> > > Wu, et al. A finite-time analysis of two time-scale actor-critic methods. Advances in Neural Information Processing Systems, 33, pp.17617-17628.

---

### Official Review · Reviewer_Pmsr · 2026-03-11

**Soundness:** 3
**Presentation:** 3
**Significance:** 2
**Originality:** 2
**Overall Recommendation:** 4
**Confidence:** 3

**Summary:**

This paper characterizes non-uniform learning rates where the last layer weights $w$ are updated much faster than the internal/"body" parameters $M$. They show that this can be viewed as an alternating optimization problem on a Stackleberg objective. Under the assumption that the objective is strongly convex in $w$, and that the induced Stackleberg objective $\Phi(M) = f(M,w^\star(M))$ is weakly convex, they can establish convergence guarantees. They provide some examples where an original objective is non-convex in the joint parameters $(M,w)$ but the Stackleberg objective becomes convex and that the induced objective can have higher curvature. They provide examples including two layer neural networks on regression, classification and TD learning settings.

**Compliance With Llm Reviewing Policy:**

Affirmed.

**Final Justification:**

The authors clarified the convexity assumptions and their choice to use RMSprop in experiments. They also provided information on the connection to Hodgkinson et al. I am still slightly concerns about (1) regimes where unbalanced learning rates lead to suboptimal behavior theoretically as it can degenerate into kernel methods (tasks such as sparse polynomials in high dimension could be problematic) and (2) the potential confounds related to global learning rates when comparing unbalanced and balanced LRs, so I maintain my weak accept stance.

**Key Questions For Authors:**

1. What would happen if the training loss can be purely minimized by optimizing $w$, but this could lead to poor generalization as the first layer features are not optimized? In this regime the network could behave more like a random feature model, which are known to achieve suboptimal generalization in certain learning tasks such as single-index models. This paper focuses primarily on optimization, but could this technology say anything about generalization when training with unbalanced learning rates?

2. Is there a possible connection between the Stackleberg objective and other approaches that redefine the optimization in terms of the last layer feature kernel $\Phi$ and the output weights directly. For example, there is the work of Hogdkinson, Wang and Mahoney https://arxiv.org/abs/2506.03470 or various works in the Bayesian NN literature https://arxiv.org/abs/2502.07998 and https://arxiv.org/abs/2108.13097.

3. Is it a fair comparison to compare training all layers with learning rate $\alpha$ and training all body layers with $\alpha$ and last layer with $c \times \alpha$ where $c > 1$ (such as Figure 2). Wouldn't the fair comparison be to compare losses at (1) optimal global LR $\alpha_\star$ and (2) optimal learning rate for pairs $(\alpha_\star , \beta_\star)$? The extent to which optimization is improved in the decoupled regime could then be measured from the difference in optimal losses. This would control for an overall speedup that could arise from some parameters updating faster than the base rate $\alpha$.

**Limitations:**

The authors could include in a limitations section the fact that the assumption of weak convexity of the Stackleberg objective has not been proven for arbitrary networks, but rather assumed in the analysis.

**Strengths And Weaknesses:**

***Strengths***

**Interesting Approach to Analyze Two-Timescale Training**
This paper provides some interesting perspectives on non-uniform learning rates by showing that the limit of an infinite timescale separation leads to a reformulation of optimization as an alternating minimization algorithm on an induced Stackleberg objective. The authors show some promising initial results, including that the induced objective can be more convex and sharper than the original objective. The analysis predicts that there are generically different rates of convergence for the joint optimization ( $O(k^{-2/5})$)  and the alternating optimization of the Stackelberg objective ( $O(k^{-2/3})$). The proof techniques, which allow for non-smooth activations, are interesting and involve Moreau envelope theory and Lyapunov functions to establish convergence.

**Connecting to Experiments**
I appreciate the efforts of the authors to examine non-uniform learning rates in several experimental settings including regression problems, classificaiton problems, and temporal difference reinforcement learning. For a theory paper, I think this enhances the potential impact of the paper.

***Weaknesses***

**Generality of Results Unclear**

A key weakness of the present results is that it is unclear *when* or rather under what conditions a deep neural network trained with unbalanced learning rates can yield a convex Stackleberg objective. Thus the assumption 3.3 may not be justified in the general case. For example, two layer networks are examined

**Some gaps between theory and experiments**

The theory is derived for SGD with decaying step sizes but many experiments use RMS prop or other optimization algorithms. The theory also mainly covers two layer networks and it is unclear whether the Stackleberg objective would be convex for depth 3 or greater.

**Concern about Fair Experimental Comparisons**

In experiments, the authors frequently compare training with identical learning rates and training with a higher last layer learning rate. I have some concerns about whether this comparison makes sense (see question 3 below).

---

> ### Author Rebuttal · Authors · 2026-03-30
>
> We thank the reviewer for the valuable feedback and thoughtful questions. We provide comment-by-comment responses below, and have made the corresponding revisions in the paper. Due to the space limit, we cannot respond to all questions. We would greatly appreciate it if the reviewer could kindly refer to our additional responses, which we will post once the author–reviewer discussion period begins.
>
> 1. **Reviewer's comment:**
>
> >A key weakness of the present results is...
>
> >The analysis predicts that there are generically different rates of convergence for the joint optimization $(\mathcal{O}(k^{-2/5}))$ and the alternating optimization of the Stackelberg objective $(\mathcal{O}(k^{-2/3}))$.
>
> **Response:**
> We clarify a possible misunderstanding here. The first comment in fact is about two separate questions.
>
> First, we do not assume that $\Phi$ is **strongly convex**. When $\Phi$ is not (strongly) convex, the convergence of SGD under non-uniform learning rates is established in a local, first-order stationarity sense in Theorem 3.6. If the problem is structured such that $\Phi$ is strongly convex, the bound in Theorem 3.7 applies and we can get a global convergence guarantee with a faster rate. While we show that there exist problem instances having a non-convex joint objective but a strongly convex reduced Stackelberg objective $\Phi$, it is true that we do not have the tools to easily test whether this favorable structure arises for a given problem instance. Our result in Lemma 4.1 is thus intended to highlight a structural possibility, and we agree that developing practically testable conditions is an important direction for future work.
>
> Second, we note that Assumption 3.3 (**weak convexity** of the Stackelberg objective) is significantly milder than (strong) convexity and is standard in the analysis of non-smooth optimization. In particular, weak convexity holds whenever the objective is smooth, and can also be established in many non-smooth settings (e.g., with ReLU activations), as discussed in Section 5.1. Importantly, we also note that the weak convexity assumption is not specific to our Stackelberg formulation. Even when analyzing standard SGD, the assumption is still needed when the objective is non-smooth.
>
> 2. **Reviewer's comment:**
>
> >The theory is derived for SGD with decaying step sizes but many experiments use RMS prop....
>
> **Response:**
> The reviewer is correct that the optimizer used in the experiments in Section 5 is RMSProp rather than SGD. This is a purposeful choice, as we would like to test whether the advantage of non-uniform learning rates is present under optimizers commonly used in practice, beyond the specific SGD optimizer analyzed in theory. However, we note that standard SGD, rather than RMSProp, is used to produce Figure 2, which demonstrates the advantage of non-uniform learning rates in a setting that is directly aligned with our theoretical analysis. We have updated the description of experimental setup in Section 5 and the caption of Figure 2 to explicitly clarify this distinction and avoid potential ambiguity.
>
> Regarding the network depth, we clarify that our theoretical results apply to neural networks of any depth. When there are more than two layers, $M$ includes the parameters of all body layers and $w$ includes the parameters of the last layer. We occasionally focus on two-layer networks in the paper to simplify the presentation. Upon revisiting the paper, we realize that certain texts may have given the impression that our results are restricted to the two-layer setting. We have revised such statements.
>
> Finally, as discussed in our response above, our analysis does not require the Stackelberg objective to be convex. The assumption needed is weak convexity, which is a mild and standard assumption.
>
> 3. **Reviewer's comment:**
>
> >What would happen if the training loss can be purely minimized by optimizing $w$...
>
> **Response:**
> We thank the reviewer for this insightful question. We agree that if the training loss can be largely minimized by optimizing the final layer $w$ alone, the body layers may behave similarly to a random feature model, which may lead to suboptimal generalization.
>
> At the same time, we note that this issue is not inherently specific to non-uniform learning rates. Since the Stackelberg objective and the original joint objective share the same set of global optima, the issue may also arise when training with uniform learning rates.
>
> That said, we agree that the Stackelberg formulation induces a different optimization landscape, and it is not yet well understood how this affects the tendency to learn rich representations versus behave like a random feature model. It is possible that non-uniform learning rates may either mitigate or exacerbate the issue, and we cannot make a definitive claim at this point. We thank the reviewer again for raising this question and believe that it is an important question deserving further investigation.

---

> > ### Author Rebuttal · Reviewer_Pmsr · 2026-04-04
> >
> > I appreciate the response from the authors. I will maintain my score

---

> > > ### Author Response · Authors · 2026-04-06
> > >
> > > We thank the reviewer for the recognition. Please find below our response to the rest of your questions that we were not able to post due to the character limit.
> > >
> > > 4. **Reviewer's comment:**
> > >
> > > > Is there a possible connection between the Stackleberg objective and other approaches that redefine the optimization in terms of the last layer feature kernel $\Phi$ and the output weights directly. For example, there is the work of Hogdkinson, Wang and Mahoney https://arxiv.org/abs/2506.03470 or various works in the Bayesian NN literature https://arxiv.org/abs/2502.07998 and https://arxiv.org/abs/2108.13097.
> > >
> > > **Response:**
> > > We thank the reviewer for pointing out these relevant connections, especially Hodgkinson et al. (2025). We admit that we are rather unfamiliar with the subjects studied in these works, and our response below is our best educated attempt upon reading the papers.
> > >
> > > There are indeed similarities between our Stackelberg formulation and the approach taken by Hodgkinson et al. (2025). In particular, both approaches "split an arbitrary learning task into a subproblem of finding optimal model coefficients for a prescribed set of features, and a main problem of identifying those features", and $q(\Phi)$ introduced in Proposition 3 of Hodgkinson et al. (2025) takes a similar form to $\Phi(M)$ we define in Eq.(3).
> > >
> > > At the same time, there are important differences. Hodgkinson et al. (2025) expresses, estimates, and optimizes $q(\Phi)$ directly in terms of distributions over functions and probability densities, using the particular problem structure. In contrast, our work studies optimizing $\Phi(M)$ through gradient samples of $f$, taking a more problem-structure-agnostic approach.
> > >
> > > 5. **Reviewer's comment:**
> > >
> > > >Is it a fair comparison to compare training all layers with learning rate $\alpha$ and training all body layers with $\alpha$ and last layer with $c\times\alpha$ where
> > > $c>1$ (such as Figure 2). Wouldn't the fair comparison be to compare losses at (1) optimal global LR $\alpha_\star$ and (2) optimal learning rate for pairs $(\alpha_\star,\beta_\star)$? The extent to which optimization is improved in the decoupled regime could then be measured from the difference in optimal losses. This would control for an overall speedup that could arise from some parameters updating faster than the base rate $\alpha$.
> > >
> > > **Response:**
> > > We thank the reviewer for this thoughtful suggestions, which we fully agree with in principle. The practical challenge, however, lies in reliably identifying such optimal learning rates. While we have conducted coarse grid search for the best uniform and non-uniform learning rates in the experiments, we do not know if what we find are truly the optimal choices. To make the most fair comparison to our ability, we compare non-uniform learning rates (i.e. $\alpha$ for the body layers, $c\times\alpha$ for the final layer) with two alternatives: 1) uniform learning rate $\alpha$ for all layers, which we refer to as "Uniform Small" in Figures 3-5, and 2) uniform learning rate $c\times\alpha$ for all layers, which we refer to as "Uniform Large". The performance improvement of non-uniform learning rates over the second baseline rules out the possibility that the gain is solely due to having all parameters updated with a larger rate.

---

### Official Review · Reviewer_3MTE · 2026-03-12

**Soundness:** 3
**Presentation:** 3
**Significance:** 3
**Originality:** 2
**Overall Recommendation:** 5
**Confidence:** 3

**Summary:**

The work proposes a reformulation of gradient descent for deep networks with layer-wise learning rates (specifically, a learning rate for the body of the network, and one for the head) as a two-step Stackelberg objective. It derives convergence rates for this reformulation.

**Compliance With Llm Reviewing Policy:**

Affirmed.

**Final Justification:**

I find the paper interesting, with novel and sound theoretical contributions. I raised a novelty concern, which was addressed during the rebuttal. This led me to raise my score. Thus, I believe the paper deserves acceptance.

**Key Questions For Authors:**

1. This work focuses on non-uniform learning rates between the final layer (the “head”), and the previous layers (the “body”). I understand that this is sometimes the case in practice, and that this justifies the strong-convexity assumption in $w$ (Assumption 3.1). Yet, I wonder what can be said about the general case, where each layer in a deep network is assigned a different learning rate.

2. Do you envision any way to generalize the theory to this setting (e.g., via a “multi-level” iterated objective)?

**Limitations:**

Yes.

**Strengths And Weaknesses:**

Strengths:

1. The paper is remarkably well written. The mathematical formalism is concise and precise, with coherent notation.

2. The main theoretical results (Stackelberg reformulation of layer-wise learning rates) are interesting, with sound and elaborated proofs.

3. The developed theory deals with non-differentiability (which is relevant, e.g., for ReLU networks), by carefully leveraging Clarke's subdifferential.

I appreciated the range of applications discussed in Section 5, which includes regression, classification, and reinforcement learning.


Weaknesses:

1. In my opinion, the core weakness of the work lies in its novelty. The idea of non-uniform learning rates is classical, and indeed has been explored  in deep learning in various forms. Layer-wise learning rate schedules have been proposed [1, 2, 3], and similar ideas have appeared in the meta-learning literature [2]. These works are not cited in the paper; in fact, I find the related work (Section 1.1) incomplete, lacking a discussion about the literature around layer-wise learning rates in deep learning, especially on the applied side.

[1] You et al., Large Batch Training of Convolutional Networks, 2017.
[2] Huo et al., Large Batch Training Does Not Need Warmup, 2020.
[3] Liu et al., A Layer-Wise Natural Gradient Optimizer for Training Deep Neural Networks, 2024.
[4] Park et al., Meta-Curvature, 2019.

---

> ### Author Rebuttal · Authors · 2026-03-30
>
> We thank the reviewer for the time, effort, and valuable feedback. We also deeply appreciate the recognition from the reviewer on our technical contributions and writing quality. We provide comment-by-comment responses below, and have made the corresponding revisions in the updated paper.
>
> 1. **Reviewer's comment:**
>
> >In my opinion, the core weakness of the work lies in its novelty. The idea of non-uniform learning rates is classical...
>
> **Response:**
> First, we clarify that the novelty of the work does not lie in the proposal of using non-uniform learning rates. As we discussed in the introduction, there already exist several empirical studies reporting improved training efficiency under non-uniform learning rates. However, these works do not rigorously explain the observed improvements, and our goal (and technical novelty) is to provide a principled mathematical explanation for when and why such improvements occur.
>
> Second, we sincerely thank the reviewer for suggesting the related references that we have missed in our literature review. We have added the following discussion to the paper after the third paragraph of Section 1.1.
>
> "Another notable line of work (You et al., 2017; Park et al., 2019; Huo et al., 2020; Liu et al., 2024; Hao et al., 2025) study layer-wise and coordinate-wise learning rate adjustment from a different perspective, motivated by improving optimization efficiency through gradient normalization or noise reduction in stochastic gradients. These works support the broader observation that modifying per-layer learning dynamics can significantly impact convergence behavior, although they do not explicitly connect such schemes to a Stackelberg or two-time-scale formalism. These approaches also rely on the choice of a base learning rate, which is typically set equal across layers. Our work can be complementary to such studies in the sense that it raises the question of whether the further optimization efficiency may arise from setting non-uniform base learning rates across layers in combination with gradient normalization and noise reduction mechanisms."
>
> 2. **Reviewer's comment:**
>
> >This work focuses on non-uniform learning rates between the final layer (the “head”), and the previous layers (the “body”). I understand that this is sometimes the case in practice, and that this justifies the strong-convexity assumption in $w$ (Assumption 3.1). Yet, I wonder what can be said about the general case, where each layer in a deep network is assigned a different learning rate.
>
> >Do you envision any way to generalize the theory to this setting (e.g., via a “multi-level” iterated objective)?
>
> **Response:**
> We thank the reviewer for this insightful question. In fact, we have asked ourselves the same question. There exist a few papers on multi-time-scale stochastic approximation (Shen and Chen, 2022; Zeng and Doan, 2024), a framework in which the aim is to solve a coupled system of $N$ equations simultaneously and which may model the problem of nested Stackelberg optimization for assigning a unique learning rate to each layer. However, the current theoretical understanding of multi-time-scale stochastic approximation is limited to the setting where only one equation (the equation associated with the slowest time scale) can be nonconvex, while all other equations need to satisfy certain strong convexity requirement. This requirement fails to hold for the objectives associated with intermediate layers when more than two learning rates are considered.
>
> Apart from the technical difficulty, we also find that such a formulation is less straightforward to interpret conceptually. The two-learning-rate setting studied in our paper admits an intuitive interpretation: the body layers act as a representation learner that evolves on a slower time scale, while the final layer performs task-specific adaptation on a faster time scale. In contrast, assigning a distinct learning rate to every layer would correspond to a deeply nested hierarchy of objectives, for which it is less clear what the underlying optimization principle or structural interpretation should be. That said, understanding how the theory extends to multi-level learning rates is an interesting and important future direction. We believe that progress in multi-time-scale stochastic approximation, particularly in relaxing the strong convexity assumption on intermediate levels, would be a key step towards making such an extension happen.
>
> **References**
>
> Shen, H. and Chen, T., 2022. A single-timescale analysis for stochastic approximation with multiple coupled sequences. Advances in Neural Information Processing Systems, 35, pp.17415-17429.
>
> Zeng, S. and Doan, T.T., 2024. Accelerated multi-time-scale stochastic approximation: Optimal complexity and applications in reinforcement learning and multi-agent games. arXiv preprint arXiv:2409.07767.

---

> > ### Author Rebuttal · Reviewer_3MTE · 2026-04-03
> >
> > I thank the authors for their reply. I am fully satisfied, and I will raise my score accordingly.

---

> > > ### Author Response · Authors · 2026-04-03
> > >
> > > We sincerely thank the reviewer for the positive evaluation and constructive feedback, which has greatly helped improve the clarity and presentation of the paper. We also appreciate the score increase, and wonder if the reviewer could kindly update the score in the original review above to make the AC aware.

---

### Decision · Program_Chairs · 2026-04-30

**Decision:**

Accept (regular)

**Comment:**

# Summary
This paper studies a Stackelberg/bilevel view of neural network training by splitting the network parameters into a body parameter $M$ and a head parameter $w$, and considering the Stackelberg objective $\Phi(M) = f(M, w^\*(M))$, where $w^\*(M)$ is the unique minimizer of the population loss $f(M,w)$ over $w$ for a fixed $M$. The paper analyzes a two-timescale projected SGD algorithm under strong convexity of the inner problem $f(M, \cdot)$ and weak convexity of the (nondifferentiable) outer objective $\Phi(\cdot)$, and proves convergence rates comparable to known weakly-convex bilevel results (Hong et al., 2023). If $\Phi$ is additionally strongly convex, a faster rate is also proved, which is again in line with Hong et al. (2023). The paper also includes some illustrative discussion on why the Stackelberg view may be useful for neural network training, as well as case studies on 2-layer networks for regression, classification, and temporal difference learning.

# Comments

This paper is cleanly written and easy to follow. The authors addressed many of the questions raised in the reviews, and the final evaluations by the reviewers were generally positive. Overall, I recommend acceptance.

At the same time, I think the final version should explain more clearly the technical novelty relative to the existing bilevel optimization literature. In particular, the main convergence result looks close in spirit to known results for weakly-convex bilevel optimization such as Hong et al. (2023), although the settings are not directly comparable, and the two-timescale SGD considered here is also closer to what is used in practice (modulo the projection step) than the algorithms usually studied in bilevel optimization papers. For this reason, I would encourage the authors to explain more explicitly in the main text what the main technical differences from Hong et al. (2023) are, and what should be viewed as the main optimization-theoretic contribution of the paper.

I also still have some concerns about the assumptions used in the theorems. In particular, the use of compact domains, weak convexity of the Stackelberg objective, and multiple Lipschitzness/smoothness assumptions makes the result feel somewhat far from standard neural network settings, especially for deeper ReLU networks. The authors say that their theory extends to deeper networks, but from my reading, the proof of weak convexity of the Stackelberg objective in Lemma 5.1 seems to use the 2-layer structure. Because of this, I am a bit concerned about how meaningfully this part of the theory extends to more general deep networks. It would be good if the authors could include at least one example beyond the 2-layer case, to support that the assumptions do not essentially rely on the 2-layer structure.

Another related concern is that the proof of the Lipschitzness/smoothness assumptions in Assumption 3.5 also seems to rely heavily on the bounded-domain assumption. In this case, I am not fully sure how well the theory applies to the more general unconstrained setting. In particular, for deeper ReLU networks over an unconstrained domain, it seems not difficult to check that Assumption 3.5 would not hold globally. One reviewer also raised a related question about the bounded-domain assumption during the review process. So I think the final version should include a more explicit discussion on the role and limitation of this bounded-domain assumption, especially in relation to the paper’s broader claims about deep neural networks.